# Multi-task weak supervision enables anatomically-resolved abnormality detection in whole-body FDG-PET/CT

Sabri Eyuboglu [1,4 ✉], Geoffrey Angus[1,4], Bhavik N. Patel [2], Anuj Pareek [2], Guido Davidzon [2], Jin Long[3], Jared Dunnmon [1,5] & Matthew P. Lungren[2,5]

Computational decision support systems could provide clinical value in whole-body FDG-PET/CT workflows. However, limited availability of labeled data combined with the large size of PET/CT imaging exams make it challenging to apply existing supervised machine learning systems. Leveraging recent advancements in natural language processing, we describe a weak supervision framework that extracts imperfect, yet highly granular, regional abnormality labels from free-text radiology reports. Our framework automatically labels each region in a custom ontology of anatomical regions, providing a structured profile of the pathologies in each imaging exam. Using these generated labels, we then train an attention-based, multi-task CNN architecture to detect and estimate the location of abnormalities in whole-body scans. We demonstrate empirically that our multi-task representation is critical for strong performance on rare abnormalities with limited training data. The representation also contributes to more accurate mortality prediction from imaging data, suggesting the potential utility of our framework beyond abnormality detection and location estimation.

[1] Department of Computer Science, Stanford University, Stanford, CA, USA. [2] Department of Radiology, Stanford University, Stanford, CA, USA. [3] Center for Artificial Intelligence in Medicine and Imaging, Stanford University, Stanford, CA, USA. [4]These authors contributed equally: Sabri Eyuboglu, Geoffrey Angus. [5]These authors jointly supervised this work: Jared Dunnmon, Matthew P. Lungren. ✉email: eyuboglu@stanford.edu

The availability of large, labeled data sets has fueled recent progress in machine learning for medical imaging. Breakthrough studies in chest radiograph diagnosis, skin and mammographic lesion classification, and diabetic retinopathy detection have relied on hundreds of thousands of labeled training examples[1–4]. However, for many diagnostic interpretation tasks, labeled data sets of this size are not readily available either because (1) the task includes rare diagnoses for which training examples are hard to find, and/or (2) the diagnoses are not recorded in a structured way within electronic medical records, requiring physicians to manually reinterpret exams or extract labels from free-text reports. Even if data sets are manually annotated, common changes in the underlying data distribution (e.g., scanner type, imaging protocol, post-processing techniques, patient population, or clinical classification schema) could rapidly render the models they support obsolete. Further, it is often the case that medical data sets are labeled using incomplete label ontologies, leading to undesirable variation in performance on unlabeled subsets of the data (e.g., rare disease types), a problem commonly referred to as hidden stratification[5].

These challenges are particularly apparent when working with whole-body fluorodeoxyglucose-positron emission tomography/computed tomography (FDG-PET/CT), a medical imaging modality with a critical role in the staging and treatment response assessment of cancer. FDG-PET/CT combines the high sensitivity of PET, which enables the detection of disease at picomolar concentrations, with the higher specificity of CT and thus is more accurate than either technology alone. Indeed, FDG-PET/CT demonstrates equal sensitivity and better specificity than FDG-PET alone even in the deep nodal regions of the abdomen and the mediastinum[6,7]. As a result of these improved capabilities and the clinical value they provide, there was a 7% year-over-year increase in the number of FDG-PET/CT examinations in 2018 within the United States[8]. Unfortunately, this increase in imaging demand has coincided with a reduction in both imaging reimbursements and radiologist specialist availability[9].

While machine learning could provide clinical value within FDG-PET/CT workflows, this potential remains largely untapped due to a lack of properly annotated data. While reading an FDG-PET/CT scan, a radiologist will typically record the anatomical regions with abnormal metabolic activity within a multi-paragraph report. This task, which we call anatomically-resolved abnormality detection, is clinically important in FDG-PET/CT studies. However, manually labeling a data set for anatomically-resolved abnormality detection is particularly painstaking because it requires either performing pixel-level annotations or sorting abnormalities into a hierarchy of anatomical regions[10]. Existing approaches rely on clinical experts manually segmenting and annotating several thousand training examples[11]. In practice, radiologists report abnormalities in many regions, even those in which abnormalities are exceedingly rare. Collecting a sufficient number of training examples to train machine learning models that can detect abnormalities even in these low-prevalence regions can be extremely difficult.

To address these challenges, we present a machine learning framework for training abnormality detection and location estimation models for large medical images without manual labeling or segmentation. In particular, (1) we introduce a weak-supervision framework for rapidly and systematically generating anatomically-resolved abnormality labels from radiology reports describing FDG-PET/CT findings, and (2) we combine multi-task learning with task-specific spatial attention mechanisms to achieve high levels of automated abnormality detection and location estimation performance even on rare FDG-PET/CT diagnoses for which we have little data. Our framework enables anatomically-resolved abnormality detection in FDG-PET/CT scans, and models based on this proposed framework could potentially provide clinical value in two ways: (1) by automatically screening negatives and facilitating the prioritization of exams and (2) by enabling future diagnostic aids that could help radiologists localize abnormalities within the scan, flag rare abnormalities, and quickly draft reports.

We first curate a data set composed of 8144 FDG-PET/CT scans, their accompanying reports, and prospective normal-versus-abnormal clinical summary codes recorded by the radiologist at the time of interpretation, similar to those reported by Dunnmon et al.[12] We leverage a custom ontology of 94 anatomical regions (e.g., left lung, liver, inguinal lymph nodes) to train an expressive language model that extracts from each report the anatomical regions with abnormal FDG uptake. Using the language model predictions as weak labels in the style of ref. [13], we then train a 3D multi-task convolutional neural network (CNN) to detect abnormal FDG uptake, leveraging the FDG-PET and CT modalities jointly as input. The model detects and estimates the location of abnormalities by making binary abnormality predictions for a comprehensive set of 26 anatomical regions in which disease was present in at least 10% of the samples of our data set. We evaluate our approach on a held-out data set of 814 FDG-PET/CT scans manually annotated by radiologists. While resource-intensive, this fine-grained manual labeling of the test set helps mitigate hidden stratification effects in our analysis[5].

We find that models trained with our multi-task weak supervision framework can detect lymph node, lung, and liver abnormalities—three of the pathologies with the highest prevalence in our data set—with median areas under the ROC curve of 87%, 85%, and 92%, respectively. In a clinical screening setting, sorting a worklist by our models' predictions would enable a team to achieve sensitivities of 95%, 98%, and 96%, respectively, after only reading the first three-quarters of exams.

We perform a comprehensive set of ablation studies to assess the utility of our multi-task weak supervision framework. In one ablation, we find that using a language model-based labeler, as opposed to a rule-based one like that of ref. [14], improves the correctness of our weak labels. In another, we show that the weak labels generated by our framework can be used to train models that are statistically superior to fully supervised models trained on small, hand-labeled data sets. Finally, we demonstrate that performance continues to increase as additional weakly labeled training examples are added, an important result given that the marginal cost of labeling more training data is negligible in our framework while adding more hand-labeled data is costly.

We next evaluate the importance of training FDG-PET/CT scan classifiers in a massively multi-task fashion (i.e., training them to detect abnormalities in many parts of the body simultaneously), an approach described by Ratner et al.[15] We find that multi-task models produce an effective learned representation of a whole-body FDG-PET/CT scan that can facilitate training models for more difficult abnormality detection tasks. For instance, we show that as the relative prevalence of abnormality in a given region decreases, the performance gains afforded by our multi-task model over single-task models increase. Additionally, when compared to single-task models, our multi-task model improves abnormality detection in regions with particularly low abnormality prevalence in our data set; we observe a 20-point improvement in the adrenal gland (AUROC 78%), and a 22-point improvement in the pancreas (AUROC 78%), two regions with fewer than 3 abnormalities per 100 exams. We also find that multi-task learning halves the computational cost of training abnormality detection models.

We further explore whether pre-training with multi-task anatomically-resolved abnormality detection (i.e., training a model in a multi-task abnormality detection and location estimation

setting before fine-tuning on a different task) can facilitate the development of models for other tasks of clinical interest. Specifically, we find that our 3D-convolutional neural network, when pre-trained on multi-task anatomically-resolved abnormality detection, can be fine-tuned to predict mortality within a 180-day period (AUROC 76%). Although we do not argue that our image-based mortality prediction models should be used without additional clinical covariates drawn from EHR data, the predictive utility of these features suggests that they could be useful in augmenting existing mortality prediction models[16,17].

Finally, we evaluate the utility of the task-specific spatial attention mechanism that our models use to produce fine-grained representations for each anatomical region. We find that spatial attention not only improves performance, but also produces attention distributions that align well with human anatomy. These distributions are useful for model interpretability and provide additional evidence that the model is attending to the correct regions of the scan.

By bridging the feasibility gap associated with the lack of detailed hand-labeled training data in volumetric imaging, our study lays a foundation for building clinically viable machine learning models for FDG-PET/CT without burdensome labeled data requirements. While we focus on FDG-PET/CT, we expect our methods and findings to have broad applicability across machine learning on volumetric imaging modalities. Of particular note, our approach would enable rapid re-labeling and retraining of volumetric imaging models based on existing reporting, which could help to combat the negative effects of distribution shift on model performance in practice. Our multi-task weak supervision framework and attention-based modeling approach for abnormality detection could enable applications in other clinical settings where it is important to both diagnose and localize pathology in high-dimensional images.

## Results

### Combining weakly-supervised multi-task learning with spatial attention mechanisms enables automated abnormality detection and location estimation in FDG-PET/CT.

To circumvent the need for costly pixel-level annotations, we develop a weak supervision framework for abnormality detection and location estimation in large medical imaging examinations. The locations of abnormalities in FDG-PET/CT scans are buried in free-text radiology reports, so extracting labels for abnormality location estimation is nontrivial. Complex sentences like "there is a mildly FDG avid lingular pulmonary nodule, which has slightly increased in size and metabolic activity," indicate where abnormalities appear in the scan. We propose using expressive pre-trained language models fine-tuned on a small data set of hand-labeled sentences to extract abnormality location labels from the reports. Our labeling approach consists of three steps and is illustrated in Fig. 1a–d: (1) we automatically tag all mentions of anatomical regions (e.g. right lung, pancreas) in the report, (2) we use the language model to predict whether or not each tagged mention in any given sentence is abnormal, and (3) we reconcile these predictions into one large anatomical ontology where each region has a single probability of abnormality. This approach allows us to extract anatomically-resolved abnormality labels with only a small amount of hand-labeled data.

Using these probabilistic labels, we train an attention-based, multi-task 3D-convolutional neural network (CNN) to detect abnormal metabolic activity in each of the 26 anatomical regions with the highest prevalence of abnormality in our data set. Our model consists of a shared CNN encoder (Fig. 1e) that supports 26 binary classification task heads, one for each region. Each of these task heads is equipped with a task-specific soft-attention

mechanism and a linear classifier (Fig. 1f). The model is initially trained on all tasks jointly. Then, single-task instances of the model are fine-tuned for each task, a technique commonly employed in natural language processing (NLP)[18]. A detailed description of our supervision and modeling procedures can be found in the "Methods" section.

We evaluate our multi-task weak supervision framework through a series of experiments meant to (1) evaluate its potential clinical utility and (2) quantify how each of our modeling choices contributes to the performance levels we observe.

We first evaluate the potential clinical utility of our models along two different axes: accurately estimating abnormality locations and effectively screening abnormal exams. In Fig. 2a, we illustrate the capacity of weakly-supervised multi-task FDG-PET/CT models to detect abnormal metabolic activity in the 26 anatomical regions with the highest prevalence of abnormality. These models are supervised with imperfect labels generated by our weak supervision framework—they use no manually annotated image data. In 22 of the 26 regions, our model achieves a mean AUROC > 75% over five random seeds. In 10 of the regions, including the lungs, liver, and thoracic lymph nodes, we achieve a mean AUROC > 85%.

In Fig. 2b–d, we provide a deeper analysis of our model's sensitivity (true positive rate) in three clinically important regions: the chest, the liver, and the inguinal lymph nodes. The purpose of this analysis is to explore how our weakly-supervised abnormality detection and location estimation framework could be used to develop triage tools for workflow prioritization in PET-CT screening. In order to evaluate potential model efficacy in this use case, we frame the test data set as a worklist, which we sort using the predicted probability of abnormality output by our model. When we sort the predictions by decreasing probability of abnormality, if we only read the first three-quarters of exams, we can still achieve a sensitivity of 95%, 98%, and 96% in the chest, the liver, and the inguinal lymph nodes, respectively. In contrast, when reading exams in the order that they are received, we expect to achieve a sensitivity of 75% if we interpret only the first three-quarters of the worklist. This analysis helps contextualize the potential utility of worklist triage in clinical interpretation. It also is worth noting that larger weakly labeled data sets would only improve these results and would cost little to acquire. We can verify that the model is using information from the appropriate regions using 3D saliency maps, one of which is shown in Fig. 2e.

### Weak supervision reduces labeling costs for automated abnormality detection and location estimation.

Weak supervision underpins our modeling framework, as it allows us to overcome a lack of labeled data and train multi-task models for abnormality detection and location estimation. Here, we evaluate our weak supervision approach and compare it to alternatives. First, to assess the capacity of our labeling framework to accurately extract abnormality locations from radiology reports, we compare the labels generated by our framework to the hand labels manually extracted from the reports by radiologists. Figure 3a compares the performance of our language model labeling framework to that of a regular expression baseline. The regular expression baseline is a labeling procedure that, for each exam, scans the associated radiologist report for mentions of predefined anatomical regions. If there are no words suggesting a negative finding (e.g., "physiologic", "without", "unremarkable") in a sentence mentioning a region, the procedure assigns an abnormal label to that region and all of its parents in our regional ontology. This formulation is similar to other, domain-specific tools[10,14].

We observe a mean AUROC of 90.0% when evaluating weak labels emitted by our labeling framework against hand labels

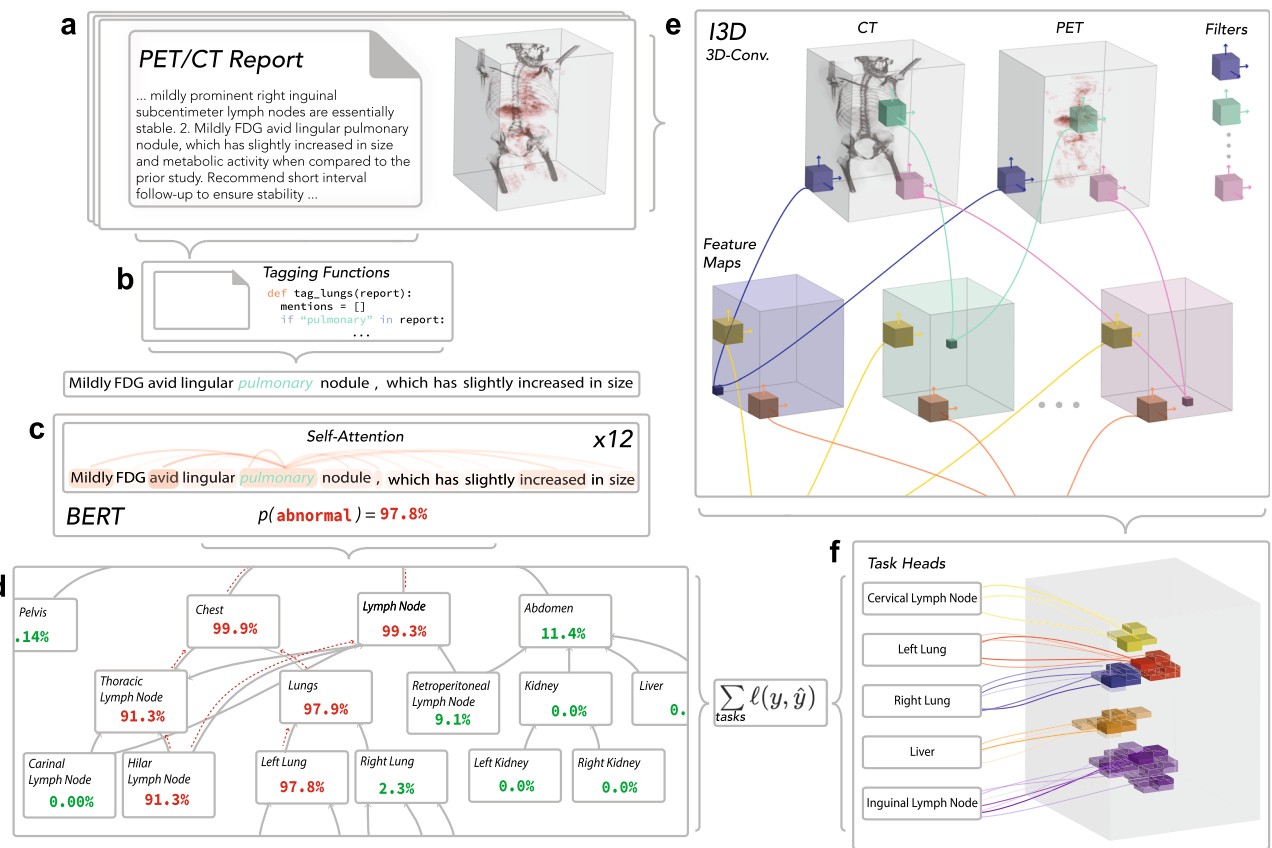

**Fig. 1 Weak supervision, multi-task learning, and spatial attention are combined to build convolutional neural networks (CNNs) for FDG-PET/CT analysis without hand-labeled training data. a** Our data set of 8144 FDG-PET/CT examinations. Each exam consists of (1) a 3D scan consisting of approximately 250 FDG-PET and CT slices and (2) a natural language, unstructured report written by the interpreting radiologist at the time of the study. Critically, there are no structured, ground truth labels for metabolic abnormalities in each anatomical region. **b** Tagging functions powered by regular expressions extract sentences that mention anatomical regions. **c** A language model predicts whether there is a metabolic abnormality in the tagged anatomical region. **d** Conflicting language model predictions are reconciled into one probability of abnormality for each region. This is done by propagating probabilities up the directed-acyclic graph of the ontology. **e** The entire whole-body PET/CT scan is encoded using a 3D-convolutional neural network. This encoding process produces a three-dimensional representation of the original scan. **f** Each task head uses an attention module to extract from the encoded scan the voxels most relevant to the anatomical region it is charged with. We apply a final linear classification layer to the weighted sum of those voxels to make a final binary prediction for each region. To train the scan model, we minimize the cross-entropy loss between the report model predictions (left) and the scan model predictions (right).

extracted from the same reports by radiologists (see "Methods" section); further, we find a 28% improvement in the mean F1-score (0.522–0.666) with respect to a regular expression baseline. With a threshold of 0.5, our model achieves a mean sensitivity of 87.0% and a mean specificity of 82.2%, outperforming the regular expression baseline, which achieved a sensitivity of 83.3% and a specificity of 66.9%. Note that we compare our labeler to the regular expression baseline using F1-score (not AUROC) because the baseline outputs binary predictions rather than probabilistic scores. The algorithm for the regular expression baseline can be found in Supplementary Note 4.

To characterize the trade-off between weak and full supervision, we train the same multi-task anatomically-resolved abnormality detection model described above on a manually labeled training data set of 400 exams. This labeling procedure, which requires annotating twenty-six anatomical regions for the presence of a hypermetabolic abnormality, took 16 board-certified radiologist hours. In comparison, our weak supervision approach required only 12 non-expert hours to label a data set of 6530 exams and could label much larger data sets with no additional manual effort. Despite the reduced labeling effort, our model trained on weak labels outperforms the model trained on hand labels in every anatomical region (Fig. 3b and

Supplementary Table 3). On average, weak supervision enables a 22-point increase in AUROC over the fully supervised model ($p = 0.0000$, paired permutation test). In the liver, for example, the weakly-supervised model detects abnormalities with an AUROC of 92.6% (91.8%, 93.4%), while the fully supervised model detects these same abnormalities with an AUROC of 57.4% (52.4%, 61.2%). In nine anatomical regions, including the neck, iliac lymph nodes, and abdomen, the fully supervised model is no better than random. Indeed, the increased data set size enabled by weak supervision leads to considerable performance gains over fully supervised approaches that use the same multi-task schema, and equivalent or greater amounts of labeling resources.

We also evaluate a hybrid approach that combines weak supervision and manual labeling. This model is trained on a mixed data set consisting of 6,530 weakly labeled exams and 400 hand-labeled exams. On average, this hybrid approach enables an improvement of 1 AUROC point over weak supervision alone—a statistically insignificant difference ($p = 0.2719$, two-sided paired permutation test). While we do see performance gains in some regions (e.g., lungs), the boost in performance is usually small.

Second, we compare our weakly-supervised whole-body abnormality detection model with its fully supervised counterpart.

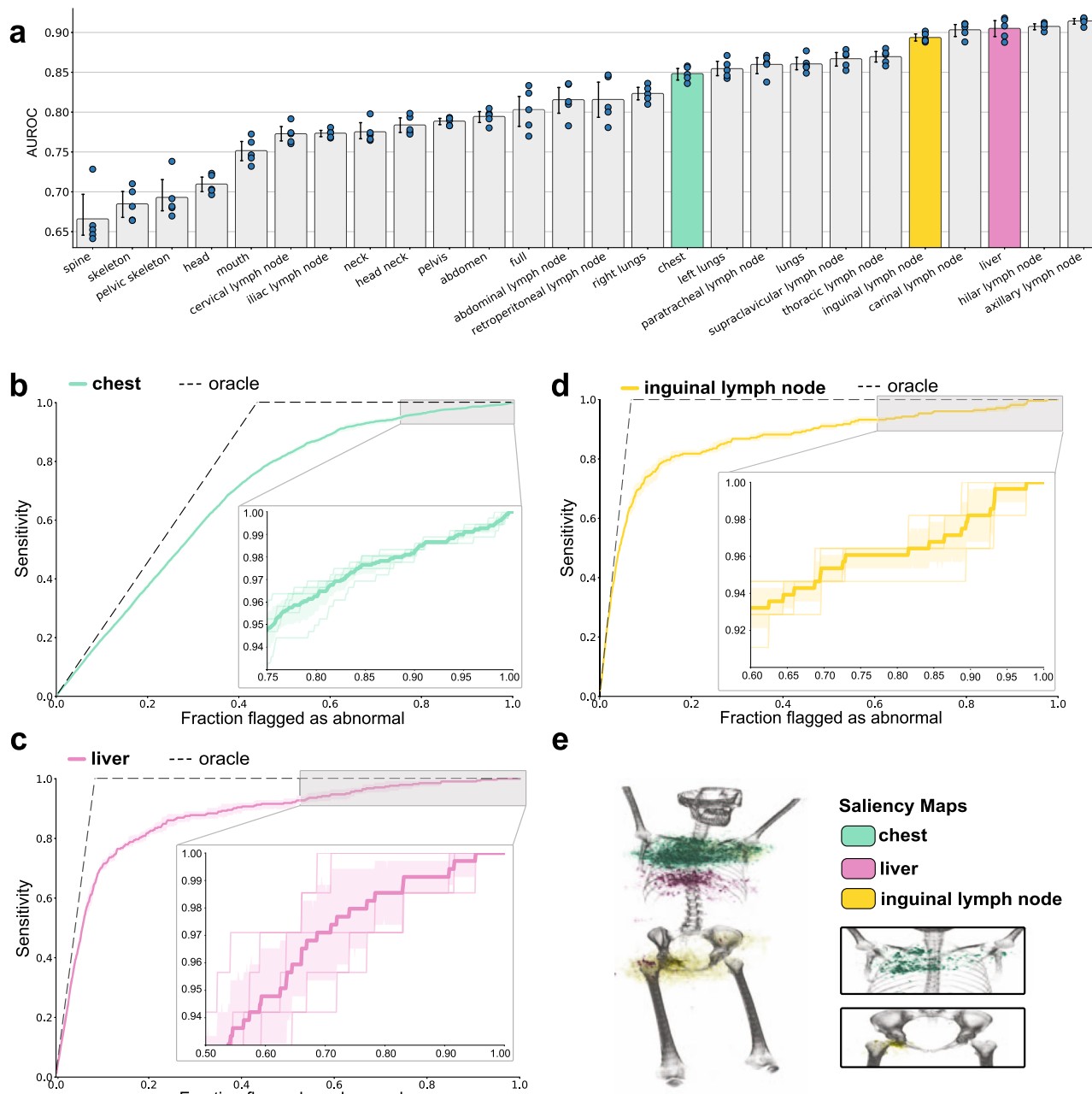

**Fig. 2 Combining weakly-supervised multi-task learning with spatial attention mechanisms enables automated abnormality detection and location estimation in FDG-PET/CT. a** Our model's predictive performance across anatomical regions. Each bar indicates the model's AUROC for detecting abnormal metabolic activity in a particular anatomical region. Confidence intervals (95%) about the mean were determined using bootstrapping over $n = 5$ random seeds. The individual AUROC result for each initialization is shown as a blue dot. Anatomical regions are sorted in order of increasing mean AUROC. **b–d** Sensitivity curve for abnormality detection in the **b** chest, **c** liver, and **d** inguinal lymph nodes. Each point on the curve indicates the sensitivity of our model at a prediction threshold where x% of the exams in our test set is flagged as abnormal. The sensitivity curve of a perfect detector (i.e., a model that ranks all abnormal examples above all normal examples) is shown in gray. The shaded region represents confidence intervals (95%) about the mean computed using bootstrapping over $n = 5$ random seeds. The sensitivity curves for each of those initializations are shown in light **b** green, **c** yellow, and **d** pink. Sensitivity curves illustrate the utility of the model in a potential screening application. With our model, clinicians could ignore around 15% of exams while maintaining 99% sensitivity in liver abnormality screening. **e** 3D saliency map for abnormality detection in the chest (green), liver (pink), and inguinal lymph nodes (yellow). Colored volumes indicate regions where small perturbations to the input scan most affect the model's prediction for chest, liver, and inguinal lymph nodes.

For each exam in our data set, we have a summary code recorded by the interpreting radiologist at the time of the exam that flags the presence of abnormality anywhere in the FDG-PET/CT scan. We use these summary codes to train a fully supervised, single-task model for whole-body abnormality detection. We also

generate analogous weak labels using our framework and train a weakly-supervised model.

In Fig. 3c, we explore how the relative performance of our weakly and fully supervised models for whole-body abnormality detection vary as we increase the number of training examples.

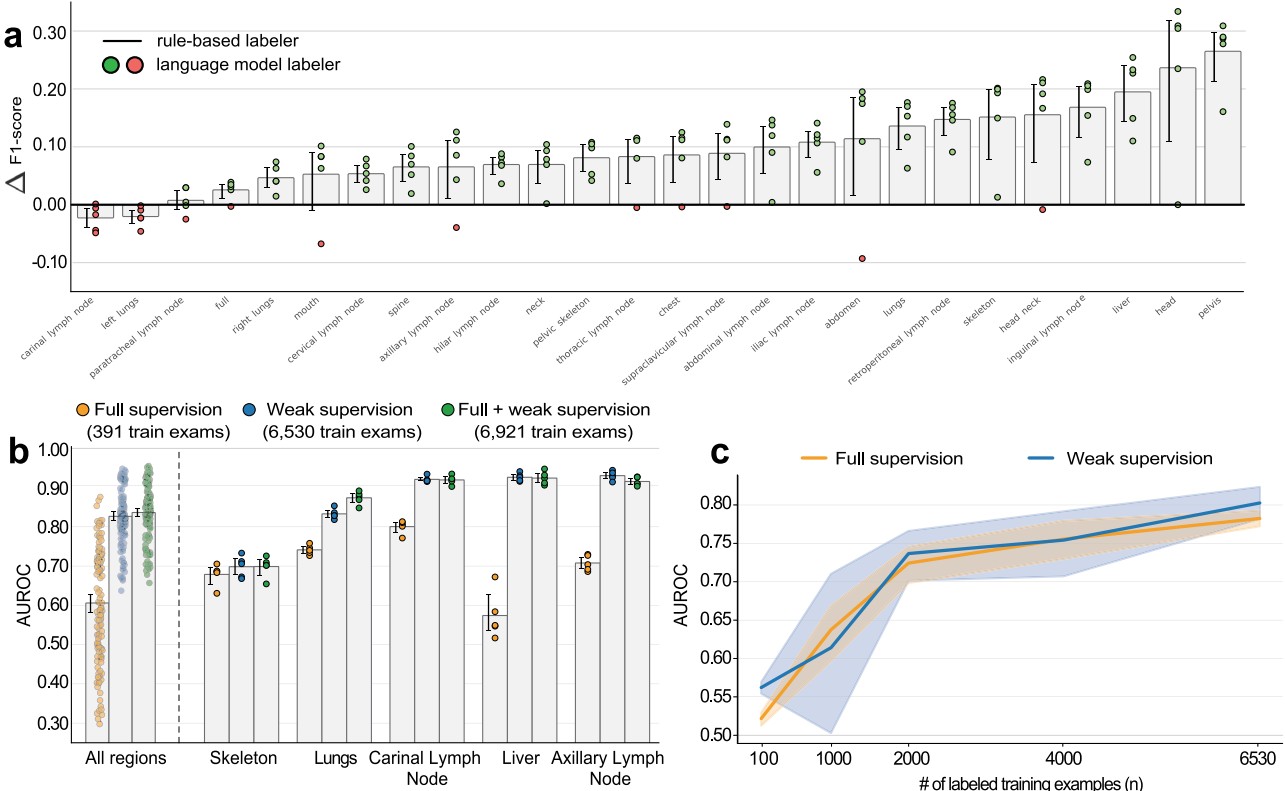

**Fig. 3 Weak supervision reduces labeling costs for automated abnormality detection and location estimation. a** Our labeling framework outperforms a regular expression baseline on 24 of the 26 regions specified in our test set. The five dots for each region represent the difference in F1-score between the baseline and our labeling framework for five different random seeds. For each region, we also show the average difference across $n = 5$ seeds and 95% confidence intervals computed via bootstrapping. **b** 26-region abnormality detection models trained on large data sets using weak supervision outperform models fully supervised on a small data set. We compare three models: (yellow, left) a fully supervised model trained using 391 manually annotated training examples, (blue, middle) a weakly-supervised model trained using 6530 automatically annotated training examples, and (green, right) a hybrid model trained using a combined data set of 6921 training examples, 391 of which are manually annotated. To the left, we show the distribution of AUROC across all 26 regions and 5 random seeds. Each point represents the AUROC on one region and one random seed. Bars represent mean AUROC across $n = 26$ regions and 5 random seeds. To the right, we compare the supervision approaches on five representative regions: skeleton, lungs, carinal lymph node, liver, and axillary lymph node. Bars represent mean AUROC across $n = 5$ seeds and 95% confidence intervals computed via bootstrapping. The numbers for all regions are provided in Supplementary Table 3. **c** Binary abnormality detection AUROC vs. the number of training data points. The performance of our weakly-supervised model is statistically equivalent to that of a traditionally supervised baseline model (trained using summary codes) ($p = 0.1933$ paired permutation test). The shape of the curve suggests that more data, which can be rapidly annotated using our framework, would lead to a continued increase in performance. The line shows the mean AUROC across $n = 5$ random seeds and shaded regions represent 95% confidence intervals computed via bootstrapping.

We find that our weakly-supervised model performs on par with the fully supervised model across a wide range of training set sizes (100, 1000, 2000, 4000, and 6530). When trained on all 6530 exams, our weakly-supervised model achieves a mean AUROC of 80.3%, which is statistically equivalent to the fully supervised model (78.3%, $p = 0.1933$, two-sided paired permutation test).

These results suggest that our weak supervision approach can dramatically reduce labeling cost while supporting accurate anatomically-resolved abnormality detection.

**Multi-task learning improves automated location estimation performance, reduces the computational cost, and facilitates mortality prediction.** Next, we show that multi-task learning enables strong performance on tasks for which we have very few positive examples and substantially reduces the computational resources required to train abnormality detection models. We also show that multi-task, anatomically-resolved abnormality detection pre-training improves patient mortality prediction performance.

Our labeling framework generates labels for 94 anatomical regions, many of which are so refined that we lack enough positive examples in our data set to train performant single-task models. In Fig. 4a, we show that multi-task learning mitigates some of the issues associated with this large class imbalance and enables us to train performant models, even on some of the regions for which we have the fewest positive examples. To demonstrate this, we chose four clinically relevant regions beyond the main 26: celiac lymph nodes (68 positive training examples, 84th most out of 94 regions), kidney (182 positives, 64th most), adrenal gland (196 positives, 60th most), and pancreas (201 positives, 58th most), and compare model performance across various types of supervision in Fig. 4b. On all four, our multi-task models substantially outperform single-task models pre-trained on the Kinetics activity detection data set[19]. In terms of mean AUROC, we see a 56% improvement (50.2–78.0%, $p = 0.0003$ paired permutation test) in detecting abnormalities in the celiac lymph nodes, a 41% improvement (63.5–89.9%, $p = 0.0022$ paired permutation test) in the kidneys, a 34% improvement (58.0–77.9%, $p = 0.0004$ paired permutation test) in the adrenal

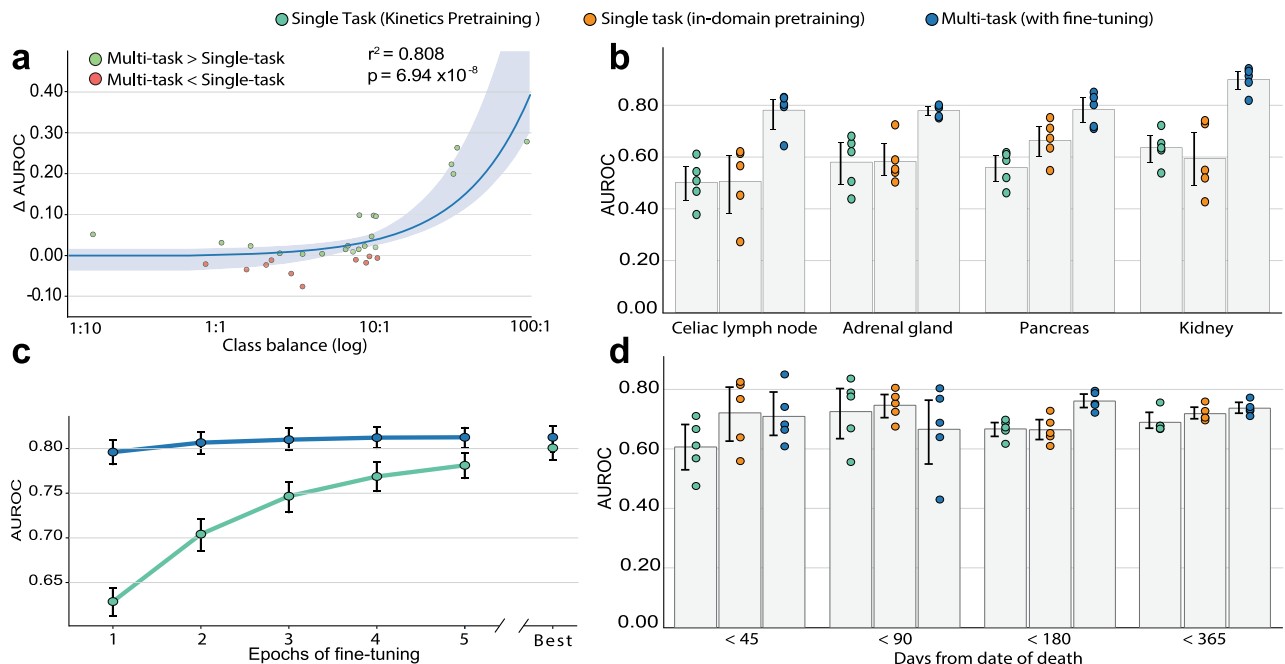

**Fig. 4 Multi-task learning improves automated location estimation performance, reduces the computational cost, and facilitates mortality prediction.**
**a** Gains from multi-task learning are the greatest in regions with severe class imbalance. For each region, the difference in abnormality detection AUROC between a multi-task model and a single-task model is plotted against the class balance for the region. If a region exhibits class balance 10:1, then for every exam with an abnormality in the region, there will be ten exams where the region is normal. The single-task and multi-task models were both trained on five different random seeds and differences in mean AUROC are plotted. We also fit a linear model to the two variables and report Pearson correlation coefficient $r^2 = 0.808$ and two-sided $p$-value $= 6.94 \times 10^{-8}$ computed using the Wald test. Shaded regions represent 95% confidence intervals computed via bootstrapping across $n = 30$ regions (e.g., pancreas, lungs). **b** Multi-task learning improves performance in regions in which hypermetabolic abnormality is rare. We fine-tune our 26-region multi-task abnormality detection model to detect abnormalities in four new regions beyond the main 26: celiac lymph nodes, the pancreas, the adrenal gland, and the kidneys. Each of these regions exhibits severe class imbalance with more than 30 positive examples for every negative example. We compare this multi-task approach (right, blue) to two other single-task approaches: (teal, left) a single-task model pre-trained on Kinetics and (orange, middle) a single-task model pre-trained on full-body abnormality detection (in-domain pre-training). Shown for each region and modeling approach is AUROC across $n = 5$ random seeds. 95% confidence intervals about the mean are computed via bootstrapping. **c** Multi-task model reduces training complexity. We fine-tune our multi-task FDG-PET/CT model in a single-task setting for each of the twenty-six core anatomical regions. We do the same with a model trained on Kinetics[41], an out-of-domain data set. The plot shows how mean AUROC across all anatomical regions improves with more epochs of fine-tuning. To the right, we show the best mean AUROC after training to convergence. Confidence intervals (95%) about the mean were determined using bootstrapping across $n = 5$ random seeds. **d** Multi-task learning improves performance on the task of predicting mortality from FDG-PET/CT scan alone. We frame mortality prediction as a binary classification task, where the model predicts whether the patient's date of death is within $x$ days of the study given only the scan as input. For $x = \{45, 90, 180, 365\}$, we show AUROC across five different random seeds. We compare three approaches: (teal, left) a single-task model pre-trained on Kinetics, (orange, middle) a single-task model pre-trained on full-body abnormality detection (in-domain pre-training), and (right, blue) multi-task abnormality detection model fine-tuned to predict patient's date of death. Confidence intervals (95%) about the mean were determined using bootstrapping across $n = 5$ random seeds.

glands, and a 40% improvement (55.9–78.2%, $p = 0.0000$ paired permutation test) in the pancreas. Additionally, we demonstrate that it is the multi-task formulation, not just in-domain pre-training, that enables these performance gains: compared to models pre-trained using weak labels on the task of binary, whole-body abnormality detection, our multi-task model achieves a 15.8% mean AUROC improvement (67.3–78.0%, $p = 0.0139$ paired permutation test) in detecting abnormalities in the celiac lymph nodes, a 25.4% improvement (71.7–89.9%, $p = 0.0180$ paired permutation test) in the kidneys, a 28.9% improvement (58.2–77.9%, $p = 0.0000$ paired permutation test) in the adrenal glands, and a 26.2% improvement (62.0–78.2%, $p = 0.0021$ paired permutation test) in the pancreas. We also compare to pre-training on summary codes and see similar improvements (Supplementary Table 7).

In Fig. 4c, we demonstrate how weakly-supervised multi-task pre-training reduces the computational resources required to fine-tune abnormality detection models until convergence. In particular, we compare the number of epochs of fine-tuning

required for convergence after pre-training on multi-task abnormality detection versus pre-training on Kinetics[19]. Our multi-task approach leads to reduced training times across all 26 anatomical regions. 68% of models pre-trained with our approach achieved their best performance within the first 4 epochs. On the other hand, only 7% of models pre-trained on Kinetics converged as quickly, while 60% required 8 or more epochs to train to convergence.

We also assess the capacity of our multi-task representation to generalize to other clinically relevant tasks. We focus on the task of using FDG-PET/CT imaging data to predict mortality. We frame mortality prediction as a binary classification task, where the model predicts whether the patient's date of death is within $x$ days of the study given only the FDG-PET/CT scan as input. In our experiments, we set $x$ to be 45, 90, 180, and 365 days. Our mortality prediction model is first pre-trained on multi-task abnormality detection in 26 anatomical regions. Then, the model is fine-tuned to predict mortality within $x$ days. We fine-tune a separate model for each threshold $x$. Our multi-task model

predicts mortality within 180 days with an AUROC of 75.8% (73.5%, 78.1%). At this threshold, multi-task pre-training enables a 14.07% improvement in mean AUROC over pre-training on Kinetics (66.5–75.8%, $p = 0.0031$ paired permutation test). Similarly, compared to pre-training on FDG-PET/CT summary codes in a single-task setting (see "Methods" section), multi-task pre-training enables a 14.6% improvement in predicting mortality within 180 days (66.2–75.8%, $p = 0.0077$ paired permutation test). Figure 4d compares our model to the two baselines at each of the four thresholds.

To understand why multi-task anatomically-resolved abnormality detection pre-training improves mortality prediction performance, we perform a traditional survival analysis that explores the relationship between abnormality detection predictions and longevity. We fit a Cox proportional hazard model on our mortality data using PET/CT abnormality detection predictions as covariates (see "Methods" section for details, Supplementary Table 18 for hazard ratios). Via the likelihood ratio test, we confirm that the fit is significantly better than a null model with only a baseline hazard and no covariates ($p = 1.7 \times 10^{-5}$, likelihood ratio test). The covariate corresponding to a hypermetabolic abnormality in the liver exhibits a statistically significant hazard ratio (Wald test). In Supplementary Fig. 1c, d, we show Kaplan–Meier curves stratified by the predictions for the liver and hilar lymph nodes.

The locations of hypermetabolic abnormalities appear to be predictive of mortality, but is this information still useful when we have access to other more readily available covariates like age or disease type? To explore this question, we fit a Cox proportional hazards model on our mortality data using anatomically-resolved abnormality detection predictions, age, indication, and exam summary codes as covariates (see "Methods" section for details, Supplementary Table 16 for hazard ratios). We also fit a nested Cox model that includes all covariates except the abnormality detection predictions (see Supplementary Table 17 for hazard ratios). We compare the out-of-sample predictive power of these Cox models by fitting on the validation set and computing Harrell's concordance index on the test set. A Cox model fit just on age, summary code, and indication achieves a concordance index of 0.597, whereas a model that incorporates abnormality location prediction achieves a concordance index of 0.636. Via the likelihood ratio test, we also show that the fit with abnormality predictions is significantly better than the fit without ($p = 7.4 \times 10^{-5}$)[20]. Indeed, the locations of hypermetabolic abnormalities seem to provide useful signals not present in other, more simply-attained covariates such as age or indication.

To evaluate the importance of each covariate, we compute the difference between the covariate's Wald $\chi^2$ statistic and the covariate's degrees of freedom as described in ref. [21] (Supplementary Fig. 1a). The two most important covariates in the model, according to this metric, are abnormality detection in the liver and the indication of lung cancer.

**Spatial attention mechanisms improve automated location estimation performance and facilitate model interpretation**. In order to train a model with the flexibility to make effective use of the granular labels produced by our framework, we incorporate an attention mechanism into each of the task heads, which learns to attend to the task-specific anatomical region. We compare a multi-task FDG-PET/CT model that uses a spatial, soft-attention mechanism in each task head (see "Methods" section) with a near-identical multi-task FDG-PET/CT model that uses sum reduction in place of soft-attention. On 22 of the 26 tasks, the soft-attention model outperforms the sum reduction model. In Fig. 5a, we show overall performance (far left), as well as region-specific

performance. Models trained with the attention mechanism see a modest empirical performance improvement over a naive sum reduction, with a 3 point increase in mean AUROC taken over all tasks and all seeds ($p = 0.0002$ paired permutation test). Additionally, our attention mechanism enables potentially useful new ways of interpreting model predictions. As shown in Fig. 5b, c, we can project the model's attention distributions onto the original scan to highlight the voxels that most informed the model's predictions. These can be used in conjunction with saliency maps, like those shown in Fig. 5d, e, to quickly locate abnormalities in the original scan, and confirm that each region-specific task head is attending to the correct part of the image.

Refer to Supplementary Tables 2–9 to see a full numerical breakdown of the results described in this section.

## Discussion

Anatomically-resolved FDG-PET/CT abnormality detection models could ultimately lower the burden on nuclear medicine specialists and improve the quality of scan interpretation. However, the lack of properly annotated data combined with the sheer size of each FDG-PET/CT exam makes it challenging to apply existing supervised machine learning systems to whole-body scans. There are on average 60 million voxels in a whole-body FDG-PET/CT examination, only a tiny fraction of which represent an abnormality. When combined with relatively high signal-to-noise ratios in PET and the wide variety of possible disease presentations, the size and complexity of each exam make it extremely challenging to train models that can estimate abnormality location in whole-body scans. Instead of operating on the full scan, existing approaches reduce the problem by only classifying a few slices at a time. While this makes for an easier machine learning task, it demands that nuclear medicine specialists painstakingly segment abnormalities on a large number of FDG-PET/CT slices[11,22]. If the scanner type, patient population, or clinical classification schema changes, training data will need to be relabeled.

From a technical perspective, we demonstrate in this work a set of repeatable techniques to rapidly build weakly-supervised, anatomically-resolved classifiers for high-dimensional volumetric imaging. The combination of multi-task learning, weak supervision, and attention mechanisms that we propose enables us to identify and estimate the locations of abnormalities in whole-body FDG-PET/CT scans with almost no hand-labeled data. This capability is critical for building machine learning systems that are amenable to routine updates in the course of clinical practice. We show that our weak supervision framework produces both whole-body abnormality detection models that are statistically similar to their fully supervised counterparts and anatomically-resolved abnormality detection models that substantially outperform those trained using a small hand-labeled data set. Moreover, we find that multi-task learning enables significant performance gains in anatomically-resolved abnormality detection, particularly on rare pathologies for which we have little training data. Finally, we find that task-specific spatial attention mechanisms improve performance and enable new ways of interpreting model predictions. We further justify the design of our framework through a series of ablation studies showing that our language model labeler improves accuracy in radiologist report parsing with respect to a rule-based system, that multi-task learning increases AUROC over comparable models and reduces train time, and that our attention mechanisms outperform simple sum reduction.

From a clinical perspective, the shortage of radiologist expertise, coupled with a sharp rise in utilization of FDG-PET/CT imaging, suggests that automation for triage and reporting tasks

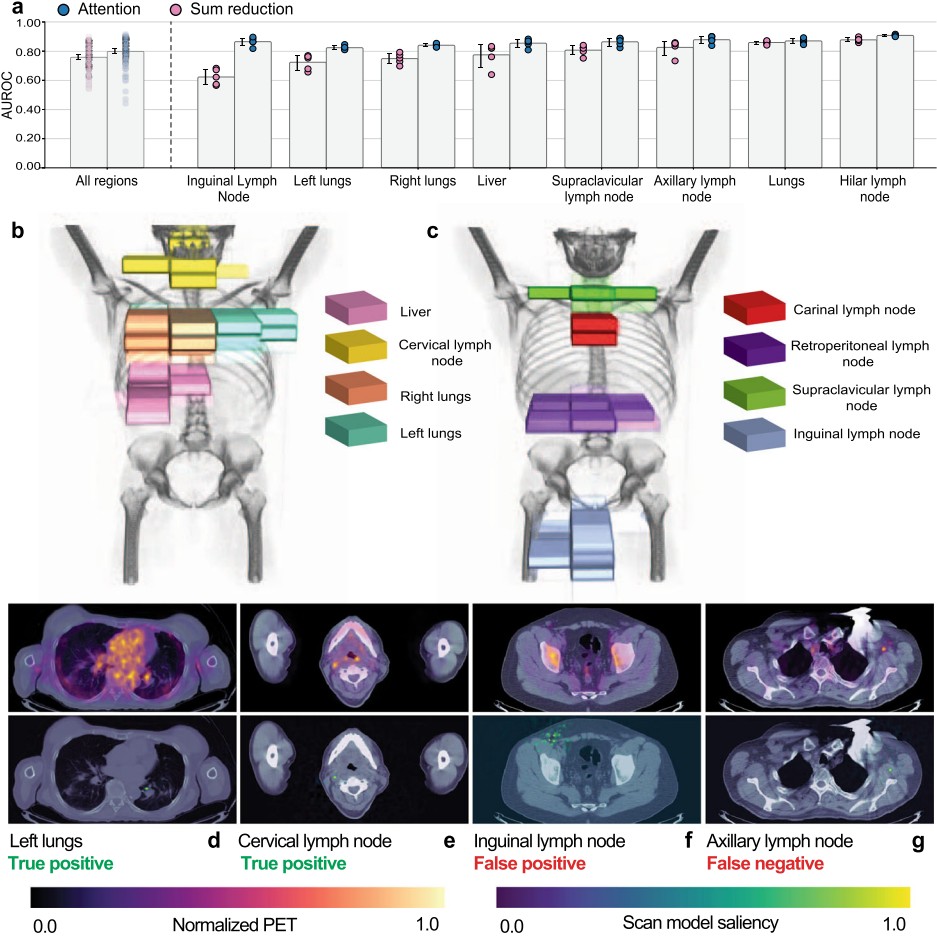

**Fig. 5 Spatial attention mechanisms improve automated detection performance and facilitate model interpretation. a** Models trained with the attention mechanism see a modest improvement over a naive sum reduction, with a 3 point increase in mean AUROC taken over all tasks and all seeds ($p = 0.0002$ paired permutation test, $n = 26$ tasks evaluated over 5 random seeds). We show overall performance (far left), as well as performance on a subset of regions. Confidence intervals (95%) about the mean were determined using bootstrapping across $n = 5$ random seeds. **b, c** The soft-attention mechanism, supervised only by weak abnormality-detection labels, learns to attend to the appropriate part of the scan. The scan encoder transforms a $2 \times 224 \times 224 \times \sim\!200$ PET-CT scan into a $1024 \times 7 \times 7 \times \sim\!33$ encoding. The soft-attention mechanism reduces this encoding to a single 1024-dimensional vector by computing a weighted sum of the voxels in the encoding (see "Methods" section). The opacity of each voxel is proportional to the weight assigned to the voxel by the soft-attention mechanism. **d–g** Two-dimensional saliency maps facilitate the interpretation of correct (**d**, **e**) and incorrect (**f**, **g**) abnormality location estimation predictions. Shown in the first row are FDG-PET uptake values overlaid over CT. FDG-PET values are normalized by subtracting the minimum value in the slice and dividing by the maximum. Additionally, in order to decrease image scatter, we clip PET values at the value of the 60th percentile pixel in the slice. Shown in the second row are saliency maps overlaid over CT. Saliency maps are produced by computing the gradient of the scan model prediction with respect to the input scan and normalizing such that the saliency for each pixel is in the range [0, 1] (see "Methods" section). The color bars for the normalized PET and saliency values are shown in the legend. **d** The model accurately detects a hypermetabolic pulmonary nodule along the medial aspect of the left lung shown in the PET (top). The model's saliency is highly concentrated in the pixels containing the nodule (bottom). **e** A right level II cervical hypermetabolic lymph node (top) is detected by the scan model. The saliency map is focused precisely on the abnormality (bottom). **f** The model detects abnormal mild FDG uptake in a subcentimeter right inguinal lymph node, around which there is a significant concentration of saliency (bottom panel). However, the clinical report does not mention this finding because, in the context of the patient's cancer, the lymph node is deemed non-specific and likely physiologic or inflammatory. Because the model and the clinical report are not congruent, this is classified as a false positive, however, the model is flagging FDG uptake that at least warrants review. **g** The model fails to detect an abnormality in a hypermetabolic left axillary lymph node of a patient with metastatic breast cancer and a pacemaker in the left chest wall (left panel). While the saliency map is concentrated on the abnormal node, the threshold for an abnormal prediction is not reached perhaps because of the overlying beam hardening artifact from the pacemaker.

may become an important part of radiologist workflows in the near future. Automated pathology location estimation could decrease the time required for interpretation and highly specific models could operationalize preliminary interpretations, draft reports, and improve workflow for radiologists and the clinical referring services. Estimating abnormality location, from a modeling standpoint, is key in that it mitigates hidden stratification or performance variability in subgroups of data due to imprecise labels[5]. In our work, we pursue abnormality location

estimation in part to provide explicit supervision over many regions of the body, in contrast to our single-task models trained on summary codes describing the whole body. Our approach ultimately leads to better performance on both whole-body abnormality detection and other downstream tasks. One of these tasks, mortality prediction, is of immense importance in palliative care. The National Palliative Care Registry estimates that less than half of all patients that need palliative care actually receive it[16]. Palliative care involves holistic and nuanced decision-making that

depends on a number of hard-to-measure variables including patient wishes, diagnosis, and available therapies. While it is beyond the scope of this study to explore how automatic interpretation of imaging data might fit into this complex decision-making process, our framework addresses some of the technical obstacles to leveraging automated analysis of imaging data in palliative care workflows.

Importantly, our approach drastically reduces the costs associated with training models for anatomically-resolved abnormality detection. In order to train the text classification model that underpins our labeling framework, only 1279 sentences (<0.3% of the sentences in our full data set) were labeled. This took fewer than a dozen non-expert person-hours. In contrast, manually labeling the 800 exams in our test set took 32 radiologist-hours—manually labeling our full training set for anatomically-resolved abnormality detection would require over 250 radiologist-hours, and this would only worsen for larger data sets. Furthermore, our weak supervision framework empowers clinical machine learning practitioners to rapidly update their labels to best fit clinical workflows and requirements[10,14]. Training multi-task models also front-loads the computational cost of training FDG-PET/CT models without compromising on model performance. Fine-tuning our multi-task models takes a fraction of the time that would be required to train an equivalent model from scratch. This further lowers the barriers to continual model refinement and retraining. Our framework provides reductions in both labeling costs and training times, bringing us closer to machine learning systems that can handle the variability and dynamism inherent in clinical practice. Further, our spatial attention mechanism could serve as another tool for model interpretation and auditing. Analysis and visualization of attention distributions, such as those shown in Fig. 5b, c, can be paired with analysis of two-dimensional and three-dimensional saliency maps (shown in Figs. 5d–g and 2e, respectively) in order to sanity check model behavior. This said the reader should note that these tools merely offer the first step towards coarse-grained model interpretation. Future work should explore whether or not they can be leveraged to reliably produce fine-grained segmentations of abnormalities.

While our results are promising in several respects, it is important to emphasize that delivering production-ready models in their final clinical form is beyond the scope of this study and that there are several additional steps that must be taken before deploying such models in clinical practice.

First, there are several approaches through which one could improve performance. As we show in Fig. 3c, increasing the size of the training data set is likely to provide additional performance improvements (other studies in clinical weak supervision have also documented this trend[23]). While our 6530-sample data set is large in FDG-PET/CT terms, it is still relatively small compared to most other medical imaging data sets in machine learning, it is drawn from a single-center, and scans were performed on devices from a single vendor using our standard institutional technique for image acquisition; each of these realities limits the generalizability of the models we present here. Since the marginal cost of adding training examples using our framework is negligible, a deployed model would ideally be trained on a more diverse data set an order of magnitude larger than ours by drawing on unlabeled scans from multiple academic centers or hospitals. This is particularly important given that performance in some regions (e.g., spine, skeleton, and pelvic skeleton) is low; increasing the number of weakly labeled training examples would be a simple and effective way to boost performance in these regions. One could also consider deploying an ensemble model comprised of multiple models, each trained from a different weight initialization, in order to account for variation inherent in the stochastic optimization procedure[24]. For example, we ensemble the five

seeds of our multi-task model to show performance gains of up to 3 AUROC points on each task, a result that can be found in Supplementary Table 20.

Second, although training anatomically-resolved abnormality detection models with our framework does not require manually annotating large amounts of training data, like with all deployed clinical machine learning frameworks, routine validation on a gold-standard, prospective data set is still critical for detecting degradation in model performance caused by shifting patient populations or changing imaging protocols. Weak labels would not suffice for these evaluations, which means that deploying models in practice will require some manual labeling of test data. This critical step will likely require significant effort: hand-labeling our test set of 800 exams took 32 radiologist-hours. That being said, if an evaluation indicates that model performance has degraded, our framework makes it easier to integrate new training data to address the issue. Note that these evaluations would be required for any automated interpretation system that would be deployed in clinical practice, and thus the requirement is not specific to the models we propose.

Third, in this study, we show how our scan model can provide a useful signal for mortality prediction. We chose to train mortality prediction models on imaging data alone not because doing so would necessarily be optimal in clinical practice, but because it highlights some of the technical challenges associated with integrating high-dimensional imaging data into mortality prediction. In practice, palliative care decisions should be based on a number of different factors, some hard to measure. We show that automated analysis of PET/CT data could provide useful signals not present in more simply-attained covariates; however, the set of covariates we considered was quite limited. Future studies should evaluate the utility of PET/CT imaging data for mortality prediction in the context of a broader set of covariates. Patients were not prospectively followed for clinical outcomes in this retrospective study design. Dates of death were acquired retrospectively and patients without a date of death were censored in our analyses. Future studies should evaluate mortality prediction models that incorporate learned representations of FDG-PET/CT data in a controlled clinical trial population with specific follow-up protocols designed for the purposes of this research question.

In conclusion, we have developed a framework for rapidly training machine learning models to detect and estimate the location of abnormalities in whole-body FDG-PET/CT scans with sparse hand-labeled data. We provide experiments justifying each of the modeling choices behind our framework—from the decision to weakly supervise models using a large, unlabeled data set to our use of multi-task learning to the inclusion of spatial attention mechanisms. We also show how we can leverage PET-CT data to provide value in mortality prediction and explore how the locations of abnormalities relate to a patient's survival function. Importantly, the techniques we've outlined can be adapted to train models for other imaging modalities where labeled data is scarce. We hope that our work can serve as a stepping stone for future research in automated analysis of FDG-PET/CT and other volumetric imaging modalities, both through its technical contributions and through its potential for future clinical impact.

## Methods

**Data set**. We use a data set of 8144 FDG-PET/CT exams from 4691 patients (Fig. 1a). The exams were administered at Stanford hospital between 2003 and 2010. Each exam includes PET and CT axially-oriented image sequences that span from the upper thigh to the base of the skull. Each FDG-PET/CT exam also includes a summary code and a free-text report written by the interpreting radiologist at the time of the examination. The summary code describes the overall patient status and takes on a value of 1, 2, 4, or 9, where 1 indicates no evidence of abnormality, 2 indicates that there are abnormalities but they were known prior to the exam, 4 indicates that there is at least one new abnormality not previously

known, and 9 indicates the presence of an abnormality that requires emergent attention. Details about the data set and the data preprocessing procedure can be found in Supplementary Notes 1 and 2, respectively. Details regarding the exam and patient metadata can be found in Supplementary Tables 12–15.

FDG-PET/CT is a combination of two different imaging modalities. The FDG-PET scan, which captures metabolic activity within the body, is composed of a sequence of $128 \times 128$ pixel images. The CT scan, which captures anatomical structure, consists of $512 \times 512$ pixel images. Figure 1a shows an example FDG-PET/CT scan, visualized such that the image frames are stacked vertically into a three-dimensional volume. The length of the image sequences used in this study ranged from 69 to 307 ($\mu = 212.39$, $\sigma = 23.77$) images.

An FDG-PET/CT report is an unstructured text document describing the clinical context of the patient's exam and the findings of the interpreting radiologist. The report typically consists of four sections: (1) the "clinical history" section, which specifies the indication of the study as well as any prior disease and/or treatment, (2) the "procedure" section, which describes the techniques used in administering the exam, (3) the "findings" section, which details clinically significant observations made in each region of the body, and (3) the "impression" section, which serves as a summary of the most significant observations made in the report[25].

This study was approved by the Stanford Institutional Review Board. The Stanford Research Repository (STARR, formerly STRIDE) is Stanford Medicine's approved resource for working with clinical data for research purposes[26]. FDG-PET/CT imaging data and radiology reports for patients in the cohort were acquired with STARR and deidentified.

The exams were split into training, validation, and test sets. The validation and test sets were sampled at random with uniform probability to match the class distribution expected in a clinical setting. The exams were split by patient, ensuring that there is no patient overlap between the training, validation, and test sets. The validation set was used to evaluate prototypes during model development, to tune hyperparameters, and for early stopping during model training. All reported metrics were computed on the test set, unless otherwise specified. For experiments comparing supervision strategies (see Fig. 3b and Supplementary Table 3), we use a hand-labeled test set of 423 exams from 235 patients, a validation set of 800 exams from 469 patients, and a training set of 6921 exams from 3987 patients, of which 391 exams from 235 patients are hand-labeled. For all other experiments, we use a validation set of 800 exams from 469 patients, a hand-labeled test set of 814 exams from 470 patients, and a training set of 6530 exams from 3752 patients.

**Weak supervision**. Recent work has established the effectiveness of weak supervision: a machine learning paradigm wherein supervised machine learning models are trained with imperfect, yet cheaply generated training labels. With weak supervision, we can reduce or even eliminate the need for costly hand-labeled data[13,27,28].

In this study, we work from within the weak supervision paradigm and implement a labeling framework that ingests radiology reports—rich, yet unstructured bodies of text describing the findings of the interpreting radiologist—and outputs probabilistic labels for each anatomical region in the FDG-PET/CT scan (e.g. lungs, liver). Our labeling framework leverages (1) a custom ontology of anatomical regions relevant to FDG-PET/CT (Fig. 1d), (2) programmatic functions that tag anatomical regions in the reports (Fig. 1b), and (3) a text classification model, which we call the report model, that determines whether the tagged regions are described as metabolically abnormal in the report (Fig. 1c). We then use the labels generated from the reports to train a large convolutional neural network, which we call the scan model, to predict abnormalities in the full FDG-PET/CT scans. Although we depend on reports to generate training labels, at test time the trained scan model can detect abnormalities in FDG-PET/CT scans without an accompanying report.

**Regional ontology**. We construct an ontology of 94 anatomical regions relevant to FDG-PET/CT. Anatomical regions include coarse, high-level regions like "chest", "abdomen", and "pelvis" as well as fine-grained regions like "left inguinal lymph node" and "upper lobe of the right lung". Our ontology is a directed-acyclic graph where nodes represent regions and edges connect regions to sub-regions. For example, edges lead from "lungs" to "left lung" and from "thoracic lymph node" to "hilar lymph node". To determine which anatomical regions to include in the ontology, we perform a systematic analysis of the region mentions in FDG-PET/CT reports. Specifically, we compute $k$-gram counts ($k = 1, 2, 3$) across all reports in our training data set. Then, if a $k$-gram refers to an anatomical region and appears at least 35 times in our data set, we add the anatomical region to the ontology. The edges connecting regions to sub-regions were added in consultation with nuclear medicine and radiology specialists (G.D. and M.L.). For a visualization of the full ontology, see Supplementary Fig. 3. Note, when training a multi-task scan model (see "Scan model training" section), we use only the 26 regions for which there is at least one positive for every nine negative examples.

**Tagging functions**. Each anatomical region in the ontology is accompanied by a set of tagging functions that search a report for mentions of the region (Fig. 1b). A tagging function could be a simple regular expression query or a complex set of rules that capture more elaborate descriptions of the region.

Given a report and a region from the ontology, tagging functions allow us to extract a list of sentences that mention that region. In our experiments, we run the tagging functions on the findings and impression sections of each report.

Note that the mention of an anatomical region does not necessarily imply that there is an abnormality in that region. In fact, the majority of mentions in our data set occur in the context of a neutral finding. Existing approaches depend on words of negation and uncertainty (e.g., "not", "no", "unlikely") to classify mentions as neutral or negative. However, compared to other modalities, the language in whole-body FDG-PET/CT reports is quite nuanced, so these approaches produce a large number of false positives (a baseline labeler that uses this approach achieves a sensitivity of 83.3%, but a specificity of only 66.9%). For example, it requires a nuanced understanding of FDG-PET/CT language to know that the sentence "intense physiologic uptake in the cerebral cortex" is describing a neutral finding.

**Report model**. Rather than rely on hard-coded rules to classify mentions, we leverage an expressive language model that can capture the complexity of FDG-PET/CT reports. Our language model, which we call in this section the report model, is trained to predict whether an anatomical region is mentioned in the context of an abnormal finding (Fig. 1c). With a trained report model, we can assign a probability of abnormality to each mention returned by the tagging functions.

Recent work in natural language processing has shown that pre-training large-scale language models on lengthy corpora of unlabeled text can enable strong performance on downstream tasks with relatively little labeled training data[29–32]. One such language model known as BERT (Bidirectional Encoder Representations from Transformers) learns word representations by conditioning on context to both the left and the right of the word[29]. Our report model is based on the BERT language model.

The model accepts as input one or more sentences of natural language. As a preprocessing step, we split up the sentences into a list of wordpieces[33]. A wordpiece is a sequence of a few characters that make up part or all of a word. The model treats each wordpiece as an indivisible unit. Wordpieces allow us to operate on rare, out-of-vocabulary words. For example, the sentence:

FDG uptake in subcarinal and contralateral mediastinal lymph nodes.

might be represented with the following wordpieces:

[FDG uptake in sub-carinal and contra-lateral media-stinal lymph nodes .]

Notice that rare, complex words like "subcarinal" and "contralateral" are split into wordpieces while common words like "lymph" and "uptake" are kept intact. Wordpieces are particularly useful for FDG-PET/CT reports where prefixes like "sub-" and "hyper-" often play important semantic roles.

BERT is typically used with a vocabulary of 30,000 wordpiece tokens optimized for generic text (BookCorpus and English Wikipedia)[34]. Because FDG-PET/CT reports are filled with highly specialized vocabulary, using BERT's out-of-the-box wordpiece tokens will mean splitting up many important, domain-specific words. To account for the specialized FDG-PET/CT text, we use a greedy algorithm to find the set of 3000 wordpieces that minimize the number of tokens required to reconstruct the reports in our training data set[33,35]. Of these 3000, 1675 are already in the BERT vocabulary. We add the remaining 1343 wordpiece tokens to the original vocabulary by replacing the "unusedX" and non-ASCII tokens provided in the BERT implementation. This allows us to leverage the initial BERT weights while also introducing domain-specific tokens. Note, it is important to exclude any testing data when generating wordpieces to ensure that evaluations on the test set provide a fair estimate of generalization error. We use Google's SentencePiece library to generate these wordpiece tokens.

We can formalize our report model as a function $\mathcal{G}$ (parameterized by $\theta_{\mathcal{G}}$) that maps a sequence of wordpiece tokens $(x_1, x_2, \ldots x_n)$ to an equal-length sequence of hidden representations $(\mathbf{z}_1, \mathbf{z}_2, \ldots \mathbf{z}_n)$,

$$\mathcal{G}(x_1, x_2 \ldots x_n; \theta_{\mathcal{G}}) = (\mathbf{z}_1, \mathbf{z}_2 \ldots \mathbf{z}_n). \quad (1)$$

The report model has a Transformer architecture, which uses self-attention to draw relations between tokens in the input. Because we use an implementation identical to the original, we refer the reader to the original Transformer manuscript for details on the architecture[36].

To predict whether or not a token $x_i$ occurs in the context of an abnormal finding, we pass its hidden representation $\mathbf{z}_i$ through a single fully-connected layer with sigmoid activation. This gives us a probability

$$P(x_i \text{ is abnormal} | (x_1, x_2 \ldots x_n)) = \sigma(\mathbf{w}^T \mathbf{z}_i + b) \quad (2)$$

where $\mathbf{w}$ is a weight vector and $b$ is a bias term.

**Report model training**. To train our report model, we need a data set of sentences that mention anatomical regions and are labeled as negative, neutral or positive. Formally, we can train the model with labeled examples $((x_1, x_2, \ldots x_n), (y_1, y_2, \ldots y_n))$ where $y_i \in \{0, 1, -\}$ takes on a value of "0" if token $x_i$ forms part of a mention

that is neutral or negative, "1" if token $x_i$ forms part of a mention that is positive, and "−" if $x_i$ is not part of any mention. For example, the sentence

$\mathbf{x}$ = [Abnormal FDG uptake in the left lung]

would be labeled

$\mathbf{y}$ = [− − − − − 1 1]

because only the words "left" and "lung" are part of the anatomical mention.

During training, we use a loss function that ignores wordpiece tokens that are not part of any anatomical mention (i.e., labeled with "−"). Formally, our optimization objective is

$$\theta_{\mathcal{G}}^* = \arg\min_{\theta_{\mathcal{G}}} \sum_{k=1}^{M} \frac{1}{n} \sum_{i=1}^{n} \mathbf{1}[y_i^{(k)} \neq -] \mathcal{L}(y_i^{(k)}, \hat{y}_i^{(k)}) \tag{3}$$

where $\mathcal{L}$ is cross-entropy loss and $\hat{y}_i = \sigma(\mathbf{w}^T \mathbf{z}_i^{(k)} + b)$, $m$ is the number of examples in the training set, and a superscript $(k)$ is a reference to the $k$th training example.

To efficiently generate a mini-data set of labeled mentions, we implement a lightweight, labeling GUI. Using it, two non-expert annotators (G.A. and S.E.) were able to label the mentions in a sample of 1279 sentences (<0.3% of the sentences in our full data set) over the course of twelve person-hours. The GUI integrates with Jupyter Notebooks and could be used to easily label data for a different task.

Prior to training, we pre-train our report model with masked language modeling (MLM) and next-sentence prediction (NSP), the two pre-training tasks proposed in the original BERT manuscript[29]. In MLM, we randomly mask wordpiece tokens in the input and task the model with recovering the masked token using just the context around the mask. To do so, we pass each hidden representations $\mathbf{z}_i$ through a MLM task head. In NSP, we feed the model two sentences and task it with predicting whether or not the first sentence preceded the second in the corpus. We take the weights from BERT$_{\text{base}}$ pre-trained on BookCorpus and English Wikipedia[29]. Then, we perform domain-specific pre-training on our training data set of FDG-PET/CT reports.

We train all of our models with an Adam optimizer and an initial learning rate of $\alpha = 0.0001$[37]. We anneal the learning rate by half every 20 epochs. We do not employ any regularization. We train the report model with a batch size of 16. In each epoch, we sample 100 mentions with replacement from the training set. We train the model for 85 epochs. We perform early stopping with AUROC on a validation set of 61 sentences. Visualizations of the BERT attention heads can be found in Supplementary Fig. 2.

**Generating labels**. Using a trained report model, we can generate labels for our training and validation data sets. For each report, we run the tagging functions from the region ontology. This gives us a list of sentences that mention anatomical regions of interest. We then tokenize each of those sentences into a sequence of wordpieces $(x_1, x_2, \dots x_n)$ and feed them through our report model. This yields a sequence of predictions $(\hat{y}_1, \hat{y}_2 \dots \hat{y}_n)$ where $\hat{y}_i$ represents the probability that token $x_i$ occurs in the context of an abnormal finding (i.e., $P(x_i \text{ is abnormal}|(x_1, x_2, \dots x_n))$, see Eq. (2)). Because mentions can span multiple tokens, we reduce multiple predictions to a single probability for the whole mention by taking the mean probability across the tokens in the mention.

Note that some anatomical regions may be mentioned more than once in a report and others not at all. To reconcile the predictions made by the report model into a single probability for each anatomical region, we leverage the regional ontology (Fig. 1d). Specifically, by propagating probabilities up the regional ontology, we collect for each region $t$ a list of probabilities $(p_1, p_2 \dots p_{n_t})$ for all mentions of $t$ and its children. We then compute the probability, assuming independence, that at least one of those mentions occurred in the context of an abnormal finding,

$$P(t \text{ is abnormal}) = 1 - \prod_{i=1}^{n_t} (1 - p_i) \tag{4}$$

The label propagation process is illustrated in Fig. 1d.

After label propagation, we are left with a single probability $\bar{y}_t = P(t \text{ is abnormal})$ for each anatomical region $t$ in the ontology. Below, we show how we can use these predictions as probabilistic labels and train a model to detect abnormalities in FDG-PET/CT scans. The class balance for each of the anatomical regions in the training, validation, and test sets can be found in Supplementary Table 1.

**Multi-task learning**. A considerable body of research has focused on using multi-task learning to reduce generalization error in computer vision and natural language machine learning models. Multi-task learning has proven particularly useful in settings where labeled training examples are scarce[38,39]. In this work, we leverage the noisy, probabilistic labels generated by the report model to train a scan model that maps a whole-body FDG-PET/CT scan to a probability of abnormality in one or more regions of our ontology. We use a simple multi-task architecture comprised of a shared encoder module $\mathcal{F}$ (parameterized by $\theta_{\mathcal{F}}$), and $T$ region-specific

decoders $\{\mathcal{D}_1, \dots, \mathcal{D}_T\}$ (parameterized by $\{\theta_{\mathcal{D}_1}, \dots, \theta_{\mathcal{D}_T}\}$)[40]. For each region $t$, the model outputs the probability that there is a metabolic abnormality in that region. We can formalize the prediction for some input scan $\mathbf{X} \in \mathbb{R}^{2 \times 224 \times 224 \times l}$ (where $l$ is the number of image slices) and region $t$ as

$$P(t \text{ is abnormal}|\mathbf{X}) = \mathcal{D}_t(\mathcal{F}(\mathbf{X}; \theta_{\mathcal{F}}); \theta_{\mathcal{D}_t}). \tag{5}$$

To train the model, we perform the following optimization

$$\theta^* = \arg\min_{\theta} \sum_{k=1}^{M} \frac{1}{T} \sum_{t=1}^{T} \mathcal{L}(\bar{y}_t^{(k)}, \hat{y}_t^{(k)}) \tag{6}$$

where $M$ is the number of samples in our data set, $\mathcal{L}$ is cross-entropy loss, $\bar{y}_t$ is the probability of abnormality output by the report model (see Eq. (4)), and $\hat{y}_t^{(k)}$ is the probability of abnormality output by the scan model (see Eq. (9)).

**Scan model**. For our shared encoder module $\mathcal{F}$, we use an Inflated Inception V1 3D CNN (I3D) pre-trained on the Kinetics data set with optical flow[41]. We remove the final classification layer so that the encoder outputs a 3-dimensional encoding of the input scan. The encoding consists of $d = 1024$ channels, each of shape $7 \times 7 \times \lceil \frac{l}{6} \rceil$, where $l$ is the number of slices in the original exam. A schematic illustration of the encoder is provided in Fig. 1e. Formally, the encoder module $\mathcal{F}(\mathbf{X}; \theta_{\mathcal{F}})$ outputs a tensor $\mathbf{A} \in \mathbb{R}^{d \times 7 \times 7 \times \lceil \frac{l}{6} \rceil}$. The encoding $\mathbf{A}$ can be viewed as a volume where each voxel is a vector $\mathbf{a}_{i,j,k} \in \mathbb{R}^d$. We visualize this encoding on the right-hand side of Fig. 1f.

Each region-specific decoder $\mathcal{D}_t$ is composed of a soft-attention mechanism and a single linear classification layer. Intuitively, the attention mechanism allows each task head to "focus" on specific regions of the scan (Fig. 1f). To perform soft-attention, we compute the dot product between each voxel $\mathbf{a}_{i,j,k} \in \mathbb{R}^d$ in the encoding and a learned weight vector $\mathbf{w} \in \mathbb{R}^d$

$$s_{i,j,k} = \mathbf{w}^T \mathbf{a}_{i,j,k} \tag{7}$$

yielding a score $s_{i,j,k}$ for each voxel. We apply softmax across the scores $\boldsymbol{\alpha} =$ Softmax($s_{i,j,k}$), and use them to compute a linear combination of all the voxels in the scan encoding

$$\mathbf{a} = \sum_{i,j,k} \alpha_{i,j,k} \mathbf{a}_{i,j,k}. \tag{8}$$

Intuitively, the larger the $\alpha_{i,j,k}$, the more attention is paid to the voxel at coordinates $(i, j, k)$. The linear combination $\mathbf{a} \in \mathbb{R}^d$ is then fed to a final linear classification layer with a sigmoid activation. Altogether, each region-specific decoder outputs a single probability of abnormality

$$P(t \text{ is abnormal}|\mathbf{A}) = \mathcal{D}_t(\mathbf{A}; \theta_{\mathcal{D}_t}). \tag{9}$$

**Scan model training**. We train the scan model with an Adam optimizer and an initial learning rate of $\alpha = 0.0001$[37]. We anneal the learning rate by half every 16 epochs. We do not employ any regularization aside from data augmentation. Due to GPU memory constraints, we train the scan model with a batch size of only 2. In each epoch, we sample 2000 exams with replacement from the training set.

Although our labeling framework generates labels for 94 anatomical regions, in practice, we find it challenging to train a multi-task model with 94 different tasks. Instead, we train a multi-task scan model on the 26 regions for which there is a prevalence of at least one positive example for every nine negative examples (i.e., a fraction of positive examples ≥10%). We estimate the prevalence using the entire data set of 8144 exams and the weak labels generated by our labeling framework. Note the remaining 68 regions in our ontology are all sub-regions of one of these main 26. The multi-task model is trained for 15 epochs. We then perform single-task fine-tuning for the 26 multi-task regions as well as four "rare" regions with class balance <3%. Through this fine-tuning step, we are able to (a) improve performance in the main 26 regions and (b) train performant models on regions beyond the 26 most prevalent (see "Results" section). We perform fine-tuning for 5 epochs. After each epoch, we evaluate the validation set using weak labels. Note, that fine-tuning typically converges after 1–2 epochs, so training for 5 epochs is not strictly necessary in practice (Fig. 4d). After fine-tuning, we use the model weights from the epoch with the highest validation AUROC.

During training, we apply two basic data-augmentation transforms to each scan. We randomly crop the image sequence to a $200 \times 200$ pixel region, then resize to $224 \times 224$ pixels, downsampling the $512 \times 512$ CT images and upsampling the $128 \times 128$ PET images using bilinear interpolation. We additionally jitter the brightness of the image sequence by adjusting brightness throughout the sequence by a factor $\gamma \sim \text{Uniform}(0.0, 0.25)$. Further details regarding the training setup can be found in Supplementary Note 3, and ablations that demonstrate the effect of upsampling the FDG-PET images and excluding the CT modality during training can be found in Supplementary Tables 10 and 11, respectively.

**Model evaluation**. Our test set of 800 exams was hand-labeled by four board-certified radiologists (G.D., B.P., A.P., M.L.) with experience ranging from 4 to 15

years. The test set was split among the radiologists such that each radiologist labeled 200 exams. The radiologists labeled each exam based only on the contents of its associated report (i.e., the actual FDG-PET/CT exam was not reinterpreted). For each of 30 anatomical regions (26 main regions plus 4 rare regions, we examine in detail in this work), the radiologists assigned a binary (i.e. {0, 1}) abnormality label. A label of 1 was assigned to a given region when there was an explicit mention of abnormal FDG uptake in that region in the findings or impression of the report. For example, if the impression contained the sentence "There is intense radiotracer uptake in ~13 × 12 mm left cervical level-ii lymph node (slice -220, SUV 6.5)," the label would be 1. A label of 0 was assigned when the impression or findings contained either an explicit mention of no abnormal FDG uptake, as in the sentence "no pathologically enlarged or hypermetabolic cervical or supraclavicular lymphadenopathy on the current study", or no mention at all of FDG activity in the region. Exams with ambiguous or uncertain wording were flagged and reviewed by a sub-specialist PET/CT radiologist with 15 years of experience (G.D.). The labeling radiologists were trained under the supervision of this sub-specialist PET/CT radiologist and were given a document providing labeling guidelines.

Unless otherwise specified, all reported metrics in results were computed using data from the test set and the hand labels described above. No hyperparameter tuning or model development was performed with the test set.

Every model in our analysis was trained with five different random initializations (i.e., random seeds). We evaluate each of the five trained models on the test set and report the mean of the five resulting scores (e.g., AUROC). We account for uncertainty in this estimate of the mean score with 95% confidence intervals computed via bootstrapping over the sample of five scores. When comparing our models to baselines (e.g., weakly-supervised vs. fully supervised), we test the null hypothesis that the sample of scores of our model comes from a distribution with the same or lower mean than the sample of scores of the other. This is done via a one-sided paired permutation test with $n = 10,000$ iterations. Note when testing whether two models are statistically equivalent, we use a two-sided paired permutation test. For $x$-day mortality prediction, we train models at four different thresholds $x$, so when comparing pre-training strategies, we run a permutation test at each threshold and apply the Bonferroni correction when interpreting $p$-values.

We evaluate the performance of our label generation framework using positive predictive value (precision), sensitivity (recall), F1-score, and area under the ROC curve (AUROC). We use F1-score to make direct performance comparisons against a regular expression baseline. We evaluate the performance of our FDG-PET/CT model using positive predictive value (precision), sensitivity (recall), F1-score, and area under the ROC curve (AUROC).

**Model interpretation.** We present three distinct ways to interpret the predictions of the FDG-PET/CT model—the three-dimensional saliency map shown in Fig. 2e, the two-dimensional saliency map shown in Fig. 5d–g, and the three-dimensional attention map shown in Fig. 5b, c. The three-dimensional saliency maps produce a high-level visualization of model behavior. They are generated using Guided Backpropagation[42]. The gradients from each of the task heads are often on very different numerical scales. In order to visualize the task gradients jointly, we pre-process the task gradients. Let $\mathbf{X}'_t \in \mathbb{R}^{2 \times 224 \times 224 \times l}$ be the gradient of scan $\mathbf{X}$ with respect to the prediction of task head $t$. We first take the absolute value of $\mathbf{X}'_t$ and then the maximum across input channels yielding $\hat{\mathbf{X}}'_t \in \mathbb{R}^{224 \times 224 \times l}$, which we can think of as representing scalar saliency scores for each voxel in the scan. We then normalize these scores by subtracting the minimum value and dividing by the maximum value, yielding $\bar{\mathbf{X}}'_t \in [0, 1]^{224 \times 224 \times l}$. We manually set clipping thresholds $\beta_1, \ldots, \beta_T$ such that for the scan gradient $\bar{\mathbf{X}}'_t$ of each task head $t$, we create a saliency map $\delta_t$:

$$\delta_t = \begin{cases} \bar{\mathbf{X}}'_t & \text{if } \bar{\mathbf{X}}'_t \geq \beta_t \\ 0, & \text{otherwise} \end{cases} \tag{10}$$

The two-dimensional saliency maps produce a slice-wise interpretation of model behavior for a single-task head $t$. We use the aforementioned normalization scheme to compute $\bar{\mathbf{X}}'_t$ and set $\beta_t = \min \bar{\mathbf{X}}'_t$ (i.e., we do not perform any clipping).

We can additionally visualize the attention scores computed per task head for a single-task. The visualization simply maps the scalar values computed in $\boldsymbol{\alpha}$ to their respective voxels in $\mathbf{a}$. An example of an attention visualization can be seen in Fig. 5b, c. For an additional t-SNE visualization of the model activations for each task head, see Supplementary Fig. 4.

**Training with summary codes.** Each exam in our data set has a summary code that was assigned by the interpreting radiologist at the time of the study. The summary code is a single-digit number, either 1, 2, 4, or 9, that indicates the degree of abnormality in the exam. A summary code of 1 indicates that there are no abnormalities anywhere in the exam and a summary code >1 indicates that there is at least one abnormality in the scan. We derive binary abnormality labels from these summary codes and train a single-task model to detect abnormalities in the full scan. This whole-body summary code model serves as a fully supervised baseline against which we can compare our weakly-supervised multi-task model.

We train our summary code model with an Adam optimizer and an initial learning rate of $\alpha = 0.0001$[37]. We anneal the learning rate by half every 16 epochs. We do not employ any regularization aside from data augmentation. Due to GPU memory constraints, we train the scan model with a batch size of only 2. In each epoch, we sample 2000 exams with replacement from the training set.

**Mortality prediction and survival analysis.** For mortality prediction and survival analysis, we use the same data set of 4926 patients that we use for the abnormality detection experiments. Dates of death for the cohort were retrieved using STARR, which integrates medical records at Stanford with the Social Security Death Index (SSDI). Of the patients in the data set, 867 (18%) had a date of death recorded in STARR. Patients were not prospectively followed for clinical outcomes in this retrospective study design. Patients without a date of death in the SSDI are still included in training, validation, and testing. For these patients, we assume survival and censor at the date we accessed the data (November 4, 2019).

To evaluate the capacity of our scan model to predict mortality from PET/CT imaging data alone, we frame mortality prediction as a binary classification task: given a patient's last PET/CT scan in the data set, predict whether the patient's date of death falls within $x$ days of the scan date. We fine-tune our scan model on this mortality prediction task by optimizing the binary cross-entropy loss over the patients in the training set. We report AUROC evaluated on the test set. The training, validation, and test splits are the same as those used for the abnormality detection experiments and include 3987, 469, and 470 patients, respectively. We fine-tune and evaluate a separate model at four different thresholds: $x = 45$, $x = 90$, $x = 180$, and $x = 365$. We compare pre-training on (1) multi-task abnormality detection, (2) single-task abnormality detection using summary codes, and (3) Kinetics[19].

We perform further analysis to explore whether the abnormality location estimates produced by the scan model contribute independently to the prediction of mortality. In this survival analysis, the response variable is the number of days between the patient's last PET/CT exam and their date of death. For patients without a date of death in the SSID, we assume survival and right censor on the date we accessed the data. The covariates used are age at the time of scanning, indication, exam summary code, and the abnormality location estimates produced by the scan model. We fit Cox proportional hazard models on all patients that were not used to train the scan model (i.e., the patients in the validation and test sets). When computing out of sample predictive power, we fit on the validation set and evaluate on the test set.

To get the abnormality location estimates, we apply our weakly-supervised, multi-task anatomically-resolved abnormality detection model to the scans in the validation and test set. We do this over five random seeds and take the median of the five scores. This gives us a single probability of abnormality $\bar{y}_t \in [0, 1]$ for each anatomical region $t$. The probabilities for the 26 anatomical regions with the highest prevalence of abnormality (the same 26 regions, we train our multi-task scan model on) are used as covariates in our Cox models.

The other covariates (age at scanning, indication, and summary code) are retrieved from metadata in the DICOM file of the FDG-PET/CT exam. We exclude one patient whose age we are unable to retrieve. Of the 42 different indications in our data set, we consider the 13 with more than two occurrences among patients with a recorded date of death in the test set. These 13 indications are breast cancer restaging, cervical cancer, colorectal cancer, head-neck cancer, lung cancer, lymphoma, ovaries, diagnostic breast cancer, diagnostic lung cancer, diagnostic head-neck cancer, regional or whole body, diagnosis lymphoma, and uncovered scan. Each patient's indication is encoded using 13 binary variables taking on the value 0 (without the indication) or 1 (with the indication). A patient with none of these 13 indications would be assigned a value of 0 for all 13 variables. In the Cox models, 0 is used as a reference group for each indication. The summary codes, which are recorded by the interpreting radiologist at the time of the study, describe the overall patient status and take on a value of 1, 2, 4, or 9, where 1 indicates no evidence of abnormality and 2, 4, and 9 indicate the presence of at least one abnormality. We binarize the summary code $s \in \{1, 2, 4, 9\}$ as $\mathbf{1}[s > 1]$ before providing it as a covariate to the Cox model.

When fitting our Cox proportional hazard models, the baseline hazard $h_0(t)$ is modeled non-parametrically using Breslow's method. We do not use a penalizer when fitting. We use the Python implementation provided by lifelines v0.25.1 and the R implementation provided by rms v6.0[43,44].

In our analysis, we fit Cox models on different subsets of the covariates described above. We fit a multivariable model using all the covariates; single variable models for each covariate separately; a multivariable model with just abnormality location estimates as covariates; and a multivariable model with just age, indication, and summary code as covariates. For each model, we report log hazard ratios with Wald 95% confidence intervals and $p$-values (Supplementary Tables 16–18).

When interpreting these hazard ratios and ranking their relative importance for the prediction, we compute the difference between the Wald $\chi^2$ value and the predictor's degrees of freedom as described in refs. [21,45]. The greater the difference, the more important the predictor is to the model. In Supplementary Fig. 1a, we plot these differences for the multivariable Cox model that uses all covariates. We use the rms R package to compute these differences[44].

To evaluate whether our abnormality location estimates predictions explain additional variation in mortality risk that simpler covariates do not, we apply likelihood ratio tests that compare two nested Cox models, one with the abnormality location predictions and the other without[20].

Furthermore, to estimate out-of-sample predictive power, we fit a Cox model on the validation set and compute Harrell's concordance index on the test set using the implementation provided by lifelines[43,46].

**Related work**. Weak supervision is a broad term used to describe techniques for training machine learning models without hand-labeled data[10]. Distant supervision is one such technique that leverages noisy labels in order to train models for a closely related task. This technique is commonly used in NLP due to frequent correlation between easily identifiable tokens and high-level semantic meaning[39,47]. In medical imaging, distant supervision in the form of text-mining has been explored for the classification of pathology in CT and FDG-PET/CT, some incorporating a label ontology similar to our own[48–51]. Ratner et al.[52] build upon the distant supervision paradigm and propose an unsupervised framework that uses generative modeling techniques to denoise labels derived from programmatic, coarse-grain labeling functions. Their framework has been used to achieve state-of-the-art performance on numerous NLP benchmarks, and has additionally proven useful in the medical imaging domain. Fries et al.[27] effectively classify aortic valve malformations in unlabeled cardiac MRI sequences. Dunnmon et al.[23] identify abnormalities in 2D chest radiographs (CXR), knee extremity radiographs, 3D head CT scans (HCT), and electroencephalography (EEG) signals.

Using trained models to label unlabeled data, which is then used to train "student" models has been immensely successful in computer vision tasks such as object detection and human key-point detection[53,54].

Multi-task learning is a training technique that has been shown to enable learning more generalizable features in some settings, particularly in settings where the number of samples is small[40]. It is common to see multi-task learning employed in medical imaging in order to improve model performance on any one task by training several related tasks simultaneously[55]. However, such approaches are often limited by the extensive labeling costs often associated with multi-task learning. Some approaches employ semi-supervised learning in order to address this bottleneck, for example through self-teaching segmentation masks[48]. However, our model requires no segmentation data and relies only on non-expert annotations for model training. Our model also incorporates an order of magnitude more tasks than most multi-task learning models, putting it in the realm of massively multi-task learning, a paradigm explored by ref. [15].

**Reporting summary**. Further information on research design is available in the Nature Research Reporting Summary linked to this article.

## Data availability
This research used data provided by STARR, Stanford Medicine Research Data Repository, a clinical data warehouse containing live Epic Clarity warehouse data from Stanford Health Care (SHC), the Stanford Childrens Hospital (SCH), the University Healthcare Alliance (UHA) and Packard Childrens Health Alliance (PCHA) clinics and other auxiliary data from 23 Hospital applications such as radiology PACS. The FDG-PET/CT scans in the validation and test sets will be made publicly available at https://aimi.stanford.edu/research/public-datasets following human review for protected health information. Until they are made publicly available, these data may be accessed by contacting the corresponding author (all requests from accredited researchers will be granted). Other individual-level data (i.e. training FDG-PET/CT scans and all reports) will not be made publicly available due to research participant privacy concerns. At any point, accredited researchers may request to access to the training FDG-PET/CT scans by contacting the corresponding author. We have also released the parameters and raw outputs of our experiments at https://github.com/seyuboglu/weakly-supervised-petct. Source data are provided with this paper.

## Code availability
A Python v3.7.3 package that includes an implementation of our weak supervision framework, preprocessing code and experiments is available at https://github.com/seyuboglu/weakly-supervised-petct[56,57].

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

## Acknowledgements

Research reported in this publication was supported by the National Library Of Medicine of the National Institutes of Health under Award Number R01LM012966. The content is solely the responsibility of the authors and does not necessarily represent the official views of the National Institutes of Health. This research used data or services provided by STARR, "Stanford Medicine Research Data Repository," a clinical data warehouse containing live Epic Clarity warehouse data from Stanford Health Care (SHC), the Stanford Children's Hospital (SCH), the University Healthcare Alliance (UHA), and Packard Children's Health Alliance (PCHA) clinics and other auxiliary data from Hospital applications such as radiology PACS. STARR is made possible by the Stanford School of Medicine Research Office The study has received a grant from General Electric (GE Healthcare, Waukesha, WI). The authors are solely responsible for the design and conduct of the study, all study analyses, the drafting and editing of the manuscript, and its final contents.

## Author contributions

M.L. conceived the initial study. S.E., G.A., G.D., J.D., and M.L. contributed ideas and experimental designs. S.E. and G.A. implemented the framework and conducted an empirical analysis of machine learning models. G.D., A.P., B.P., and M.L. annotated validation data. S.E. and J.L. performed the survival analysis. All authors contributed to writing.

## Competing interests

The authors declare no competing interests.
