## [Peer Review File · Nature Communications]

Reviewers' Comments:

Reviewer #1:

Remarks to the Author:

This paper proposed a weak supervision framework to detect abnormality from whole-body FDG-PET/CT. It first extracted 26 abnormality labels from the radiologist reports associated with each scan, and then used the multi-task learning strategy to train a CNN with the text-mined labels. Experiments on the in-house dataset with 8,251 exams show that the multi-task CNN achieved reasonable results. The paper also shows that the framework might be useful on rare disease detection, mortality prediction, and triage tool for workflow prioritization.

Clinically, I doubt that the performance of the framework is adequate to assist clinicians. First, the overall performance (Fig 2a) is probably not outstanding enough. In general, AUC of 0.5 suggests no discrimination, 0.7 to 0.8 is considered acceptable, 0.8 to 0.9 is considered excellent, and more than 0.9 is considered outstanding (Hosmer & Lemeshow (2013). Applied logistic regression. p.177). Although weak supervision learning helps improve the performance in this study, only 4 out of 26 categories' AUC are above 0.9 (outstanding). On the other hand, 11 out of 26 categories' AUC are below 0.8 (acceptable or below). I feel the results may not reach the standard of this journal.

The second example is the triage experiments (Figure 2b-d). By using the proposed method, the clinicians still need to read 75% of the whole exams while maintaining 98% sensitive in liver abnormality screening. I suspect this result is not high enough in clinical practice.

Technically, the paper claims three highlights: (1) abnormality localization, (2) multi-task, and (3) weak supervision.

Abnormality localization: In computer vision, the task of localization is to predict the object in an image as well as its boundaries. This paper uses fine-grained labels to suggest the abnormal organs but does not predict the boundaries. Hence, it tries to solve a multi-label classification problem (if I understood correctly), rather than a localization problem.

Multi-task: the paper does not compare the single-task and multi-task settings using the text-mined labels. Without such, I cannot review the contribution of multi-task learning in this study. It seems that Fig 3b compares single-task and multi-task settings. Unfortunately, Fig 3b is not discussed in the paper.

Weak supervision: The paper compares the proposed models with (1) Kinetics model and (2) summary code model. However, I feel the comparison is not fair. First, although both baseline models are fully-supervised, they are single-task models. Second, neither of them predicts 26 categories. I would like to see a comparison of models training on (1) a small annotated training set and (2) this small annotated set plus large-scaled text-mined training set.

Rare abnormality detection: while the proposed method achieved 77% of AUC, there is no baseline (i.e., a single-task model) to understand the level of difficulty of this problem. Some rare abnormalities may be easy to detect, and a few positive training examples are enough to train the model. To prove the hypothesis that the proposed framework helps overcome the lack of data, it would be important to compare the proposed model with a baseline.

The organization of the results needs to improve. The authors are recommended to highlight the most important observations in the Results section.

The organization of the figures needs to improve. It is hard for me to follow the logic between subfigures. For example, Figure 3 contains five different topics: performance of NLP models, the relation between AUC and training data size/epochs in the image models, rare disease detection,

and mortality prediction. I am not sure the whole picture the authors would like to show.

Other concerns:

Fig 3e is before Fig 3a-d

Fig 3b is not discussed in the paper.

Fig 3e. DoD is not explained in the paper.

Supplementary Figure 1b-d. The max follow-up time is 3500 days. Should it be 365 days?

It is unclear how the CoxPH model was trained to predict the overall survival probability of mortality.

"We demonstrate that it is the multi-task formulation, not just in-domain pre-training, that enables these performance gains". The comparison between multi-task on text-mined labels and single-task on summary codes is not fair. There are two variables: learning strategy and training labels. Again, a baseline of text-mined labels in a single-task setting is needed.

The authors used k-gram to add the regions to the ontology. The authors didn't explain how the edges were added.

The authors of BERT recommend using the original vocabulary to keep fine-tuning the BERT model. This is partly because the initial weights of BERT_base rely on the vocabulary. In this article, the authors used a new vocabulary (with additional 3,000 wordpiece tokens) to fine-tune the BERT model. The authors need to explain the reason, how it will affect the results, and provide the comparison.

While the authors explain the summary code of 1, they didn't explain the meaning of the codes 2, 4, and 9. Furthermore, is it a single-task model or a multi-task model?

The authors created an ontology of 96 anatomical regions, but the multi-task model can only predict 26 anatomical regions. The authors should explain how these 26 out of 96 are chosen. The authors also need to explain if the 96 regions are too refined.

The original images are 128 x 128, while the input of CNN is 224 x 224. Will upsampling from 128*128 to 224*224 introduce noise to the PET image? What technique did the authors use for the upsampling?

"over the last five years there has been a 16-fold increase in number of FDG-PET/CT examinations within the United States". Please provide a citation.

Comparing this paper with previous papers that use text-mined labels to classify medical images is necessary. For example,

1. Gong T, et al Automatic labeling and classification of brain CT images. In 2011 18th IEEE International Conference on Image Processing 2011 Sep 11 (pp. 1581-1584).
2. Singh S, et al. Deep-learning-based classification of FDG-PET data for Alzheimer's disease categories. In 13th International Conference on Medical Information Processing and Analysis 2017 Nov 17 (Vol. 10572, p. 105720J).
3. Yan K, et al. Holistic and comprehensive annotation of clinically significant findings on diverse CT images: learning from radiology reports and label ontology. CVPR 2019 (pp. 8523-8532).

Multi-task learning is widely used in medical image analysis, some related work of multi-task

learning on medical image analysis is needed. For example,

1. Khosravan N, Bagci U. Semi-Supervised Multi-Task Learning for Lung Cancer Diagnosis. Conf Proc IEEE Eng Med Biol Soc. 2018;2018:710-713. doi:10.1109/EMBC.2018.8512294
2. Moeskops, Pim, et al. "Deep learning for multi-task medical image segmentation in multiple modalities." MICCAI 2016

Reviewer #2:

Remarks to the Author:

The manuscript "Multi-task weak supervision enables automated abnormality localization in whole-body FDG-PET/CT" by Eyuboglu et al. provides a generalizable framework for automated detection of abnormalities from FDG-PET/CT scans based on complicated machine learning and natural language processing models. The framework is highly interesting and can prove very useful for generating novel computational tools to help nuclear medicine specialists. One could also imagine similar approaches being used in contexts outside PET/CT scans. The article in itself presents the methodology and results adequately, but the content is very densely written, and therefore may be difficult to understand for some readers. This is not necessarily a limitation. Balancing adequate scientific content with readability is difficult in a complex area like this.

The method presented is truly novel and a very compelling alternative to labelling PET voxels for machine learning algorithms.

We believe the manuscript could be improved by addressing the following:

- 1) Although the presented framework enables training of an automated abnormality localizer based on whole-body FDG-PET/CT scans and weak-supervised learning without the need for manually entering abnormality locations in the training data, future models still need to be properly validated. The authors used a self-labelled test dataset to validate their model, which is a requirement also for future models in other settings. To avoid the perception that no hand labelled data are needed for validation purposes, the authors should touch upon this in the discussion.
- 2) It is not entirely clear what role the validation dataset plays in the framework. Presumably, it is used for the so-called fine-tuning step, but this is not explicitly mentioned. The authors should elaborate.
- 3) The authors could improve the readability of the results, if they were more explicit about which dataset and endpoint are used when reporting AUROC, sensitivity, F1, etc. throughout the manuscript. For instance, in the sentence "Our labeling framework enables a 28% improvement in mean F1 score (52.2% to 66.6%) over this regular expression baseline and achieves a mean AUROC of 90.0%" on page 8, it is not clear whether this is based on validation data or test data, or whether this is based on the downstream scan output or the upstream labels. In fact, it would be more interesting to look at the downstream scan output, as the presented framework does not suggest to use the labeler alone.
- 4) Although a secondary endpoint, the authors propose to predict x-day mortality based on the presented framework. However, there is no mention in the manuscript of how patients were followed and how the authors handled censoring (if any). The modelling approach in this setting explained in the sentence, "Our mortality prediction model is pre-trained on multi-task abnormality detection in 26 anatomical regions and fine-tuned on a small dataset that pairs each study with a date of death" is challenging to grasp. Does "study" mean location? What does it mean to pair with date of death? And does the same model predict mortality at all four time points? As a baseline, the authors could also compare with a model, that only utilizes the patients age, as this is much easier to obtain than FDG-PET/CT scans. If age is as good a predictor, there might be limited utility in basing mortality predictions on scans.
- 5) While impressive mortality prediction, one could be concerned about using imaging alone for this purpose. Imaging does not provide the diagnosis and while the distribution of lesions may hint to final diagnosis, relying on this for palliative care decisions would be too uncertain and could risk

patients with curable disease such as lymphoma to be assigned to palliative care. We are aware that this is not proposed for direct use by the authors, but more limitations are needed for this part.

6) The authors state that: "The importance of abnormality localization pre-training in mortality prediction could be explained by the observation that patients for which the model detected at least one abnormality showed a significantly lower survival". However, this does not explicitly suggest a benefit of detection the locations of abnormality, as this is not what the authors investigate with the Cox model. Also, the fact that lesions detected associates with mortality is not so surprising. Likely, just the fact that having a PET done is predictive of death in broad population, among those with PET done – abnormalities are predictive of death etc. I would be good to include more on this part, what type of abnormalities are strongly associated with death – and when adjusting for baseline diagnosis - is the automated PET still important?

7) The authors calculate the performance metrics over five random seeds (leading to five different models?), as the training procedure has some stochastic elements to it. The authors could reflect a little on how this should be handled in everyday use. Should many models be generated and the mean (over each model) of predicted probabilities be used as the final prediction? The authors could discuss this.

8) The reliance of imaging reports for modelling is extremely interesting, but it is important to know more about the report structure. Are they following the same structure or are reports more in free text? In some countries, a report would be free text, others more checklist like. I can imagine the latter would provide better foundation for a work like the present. Could the authors show the structure of a report in the paper? Is it a fixed format?

9) Can we have more information on the patients that are examined? Cancer patients? Which types. It would be particularly useful to describe if the model can predict type of cancer and whether it is localized or not? In some cancers, having lower sensitivity for some areas, for example for mediastinal lymph nodes may matter a lot, for example in lung cancer. How does the model perform for such tasks, can this be assessed? Helping physicians where to guide biopsy in case of predicting cancer type or as an initial assessment of cancer stage (localized vs non-localized)

10) Could the authors elaborate on whether this model will be good for metabolic volume assessments etc. Now that abnormal areas are detected could volumes be determined? Is this next step?

The manuscript would also benefit from addressing these minor comments:

1) Are the generated word-pieces (vocabulary) only based on training data or the entire dataset? They should likely be derived from the training data to ensure fair validation.

2) The authors state that "(a baseline labeler that uses this approach achieves a sensitivity of 83:3%, but a specificity of only 66:9%)". Is this in validation/test data or training dataset?

3) Do the authors have an explanation as to why a the fully supervised vision performs slightly worse than the multi-task model. Do the signal get blurred when combining all anatomical regions?

4) The authors use the term "pre-trained" throughout the manuscript. It might be standard vocabulary in machine learning communities, but in this context it suggests that the model has been trained previously elsewhere and not a part of this study. The authors could make the framework more understandable if they simply wrote "trained".

5) How are the sensitivity curves of the perfect detector calculated in Figure 2b-d. The authors could add some information to the figure label.

Reviewer #3:

Remarks to the Author:

In this manuscript, the authors aimed to develop a framework to automatically detect metabolic abnormalities in whole-body FDG-PET/CT scans using a multi-task, weak supervision approach.

General comments:

- While this study has its strengths on the technical/methodological side, it lacks profound clinical considerations. All the challenges attributed to PET/CT scans in the introduction are even more true for CT- or MRI-only examinations. The PET signal is a helpful guidance for abnormality detection, in particular for small and rare findings. This is one of the main purposes to perform a PET/CT. However, a PET/CT scan consists of both modalities - each providing unique and often complementary value. Interpreting them individually is not sufficient - thus - also relying on only one of them for anomaly detection is not sufficient because critical and potentially life-threatening information (e.g. pulmonary embolism) can be missed. Therefore, the authors should not refer to the framework as a FDG-PET/CT model but rather a FDG-PET model

- Overall, I see this study as a proof of concept investigating the possibility to generate automatically labeled training data and perform anomaly detection in PET-scans. The clinical claims such as developing an "effective triage tool" or predicting mortality as "clinically relevant task" to "identify patients that could benefit from palliative care" are exaggerated and simply not supported by the data and analyses.

- The authors investigate an interesting concept that could be helpful to address the issue for generating labeled datasets for AI applications in a more efficient way. Overall, the manuscript is well written and easy to read, however, the structure and built up of the different section could be improved. Introduction is too long - especially the last 1 ½ pages should be substantially shortened. Most of it belongs to results or discussion. Results could benefit from more subheadings. Also, limitations should be summarized in a separate paragraph in the discussion.

Specific comments:

- what about abnormalities in organs/anatomical regions with a high physiological FDG-Uptake? Did the authors consider any threshold for SUV measures to distinguish between abnormal, borderline and physiological findings? For example, in the report excerpt provided in Figure 1a multiple nodes are reported but only a single, moderate SUV uptake is mentioned.

- Methods section describing setup of Figure 1 should follow labeling order of Figure 1, i.e. 1d follows 1b and 1c.

- Methods: include more information (number of patients and scans) for the training, tuning and testing datasets.

- Methods, Model evaluation: Did each of the radiologists label all 800 cases of the test set or were they split between radiologists? What was the radiologist's experience in interpreting PET/CT scans and how were the labels assigned (probability between 0-1, was there a training session before actual labelling about how specific expressions should be handled, e.g. "unlikely", "cannot be ruled out", "is possible"? Where labels only based on reading the report? How were cases handled if wording was ambiguous?) This is a crucial step that should be described in more detail as the authors themselves mention correctly that the language in PET/CT scans is "quite nuanced".

- No information of patient characteristics are provided in the entire manuscript. This is important information from a methodological as well as a clinical perspective and a major limitation of the study.

- I am missing a separate section summarizing the statistical analyses conducted. The authors mention to use cox models to estimate survival. However, this is only mentioned once in the introduction and once in Figure S1 without providing hazard ratios or any other information about model development and other variables used.

- Figure 1: For clarity - Figure 1b should follow the example given in Figure 1a. I also wonder whether the wording in 1c actually qualifies for a metabolic abnormality. Again, there is no information about the actual SUV uptake.
- Figure 2: I cannot see the sensitivity curves for the initializations in "light blue".
- Figure 4: panel d) if I am not mistaken the described lymph node is in level II on the left not level I on the right. I also wonder whether the lymph node described in panel e) is axillary or (inter-)pectoral. Please check and revise accordingly.
- Figure S1b: caption does not match order of the panels. Could be interesting to add and compare a curve for the summary codes of the original reports in the full body plot.
- Figure S2b: the assigned mean attenuation scores are missing.
- Correction of typos needed: e.g. Introduction, second paragraph; "FDG-PET demonstrates demonstrates...." or Methods first sentence "FDG FDG-PET/CT..."

Responses to Reviewer #1

R1.1 This paper proposed a weak supervision framework to detect abnormality from whole-body FDG-PET/CT. It first extracted 26 abnormality labels from the radiologist reports associated with each scan, and then used the multi-task learning strategy to train a CNN with the text-mined labels. Experiments on the in-house dataset with 8,251 exams show that the multi-task CNN achieved reasonable results. The paper also shows that the framework might be useful on rare disease detection, mortality prediction, and triage tool for workflow prioritization.

We appreciate that the reviewer has understood the key contributions of our work and taken the time to provide us with useful feedback.

R1.2 Clinically, I doubt that the performance of the framework is adequate to assist clinicians. First, the overall performance (Fig 2a) is probably not outstanding enough. In general, AUC of 0.5 suggests no discrimination, 0.7 to 0.8 is considered acceptable, 0.8 to 0.9 is considered excellent, and more than 0.9 is considered outstanding (Hosmer & Lemeshow (2013). Applied logistic regression. p.177). Although weak supervision learning helps improve the performance in this study, only 4 out of 26 categories' AUC are above 0.9 (outstanding). On the other hand, 11 out of 26 categories' AUC are below 0.8 (acceptable or below). I feel the results may not reach the standard of this journal.

We appreciate the reviewers' question. As the reviewer points out, performance, as measured by AUROC, does vary between regions; for example, in four regions AUROC exceeds 0.9 while in four others AUROC dips below 0.75. Such variation can be attributed to differences in the inherent difficulty of detecting abnormalities in the region, the relative prevalence of different disease types in the dataset, and other factors. However, we respectfully push back on the idea that the AUC values presented in this work are not of sufficient merit for several reasons.

First, the clinical merit of these models is highly dependent on operating point. There are often cases where models with relatively low AUC values – for instance, in situations where very high sensitivity is required, but nearly any level of specificity would be helpful – provide practical value. Abnormality localization on full-body FDG-PET/CT, for instance, could be described as this type of application, as it is highly labor-intensive, and the alternative to using the proposed model is that the clinician must manually read the entire scan with no additional information about where the malignancy may lie. In this context, there is not a great deal of practical difference between an ROC-AUC value of 0.8 (which 11 of our 26 categories fall below) vs. an ROC-AUC value of 0.75 (which only 4 of our categories fall below) in our study. Further, the fact that our model does achieve a high AUC on a large number of these categories implies that it could be particularly useful for those specific pathologies.

Second, we do not believe that it reduces the merit of our work to report that our method works better for some diseases than for others; rather, we believe this is a practice worthy of responsible science.

Third, we argue that our study – which uses a large FDG-PET/CT dataset with respect to those reported in the literature, but not with respect to those collected across clinical medicine – does not represent the endpoint of how well FDG-PET/CT models of this type could perform, but rather provides an important proof-of-concept. Indeed, though we used a relatively small training dataset of 6,530 exams, the results we present in Figure 3c suggest that increasing the size of the training dataset will likely lead to additional performance improvements, in alignment with recently published work on cross-modal weak supervision in medicine [26]. Since the marginal cost of adding weakly labeled training examples using our framework is negligible, the results of this study should provide ample motivation for others – for instance, those in large health-care systems with access to tens or hundreds of thousands of FDG-PET/CT scans – to build upon our work by applying our approach to larger datasets and building models that could be even more useful in the clinic. In this context, we argue that in addition to our compelling empirical results, the detailed ablations we perform on each part of our proposed framework (weak supervision from text reports, multi-task learning, and task-specific spatial attention mechanisms) provide important information to potential practitioners who would ultimately translate our approach to a clinical setting.

R1.3 The second example is the triage experiments (Figure 2b-d). By using the proposed method, the clinicians still need to read 75% of the whole exams while maintaining 98% sensitive in liver abnormality screening. I suspect this result is not high enough in clinical practice.

As we argue above, we would push back on this for two reasons. First, automated triage systems are rarely, if ever, deployed for FDG-PET/CT; even an approach that rules out 25% of scans would relieve substantial burden on clinicians. Second, from a novelty perspective, as far as we are aware the full-body abnormality detection results presented in this paper are the first of their kind on FDG-PET/CT. Even though we use the largest FDG-PET/CT dataset of which we are aware for this task, our results (e.g. Fig. 3b) indicate that given access to additional data, we could rapidly improve these results further using our multi-task weak supervision approach.

R1.4 Technically, the paper claims three highlights: (1) abnormality localization, (2) multi-task, and (3) weak supervision.

We appreciate that the reviewer has recognized the main technical contributions of our paper. We also reinforce that we apply these techniques in concert on the largest FDG-PET/CT dataset of which we are aware to demonstrate automated PET-CT classification and localization capability using deep learning for the first time. Below are

recent studies in the domain, all of which use less than half the data leveraged in this work, most substantially less.

Kawakami, Masashi, et al. "Evaluation of Automatic Detection of Abnormal Uptake by Deep Learning and Combination Technique in FDG-PET Images." *Journal of Nuclear Medicine* 61.supplement 1 (2020): 3007-3007.

Komori, Seisaku, et al. "Image-based deep-learning prediction of future FDG PET patterns in aging and dementia." *Journal of Nuclear Medicine* 60.supplement 1 (2019): 1211-1211.

Peng, Hao, et al. "Prognostic value of deep learning PET/CT-based radiomics: potential role for future individual induction chemotherapy in advanced nasopharyngeal carcinoma." *Clinical Cancer Research* 25.14 (2019): 4271-4279.

Schwytzer, M., et al. "Automated detection of lung cancer at ultralow dose PET/CT by deep neural networks-Initial results." *Lung Cancer (Amsterdam, Netherlands)* 126 (2018): 170-173.

R1.5 Abnormality localization: In computer vision, the task of localization is to predict the object in an image as well as its boundaries. This paper uses fine-grained labels to suggest the abnormal organs but does not predict the boundaries. Hence, it tries to solve a multi-label classification problem (if I understood correctly), rather than a localization problem.

We appreciate that the reviewer has taken the time to restate this aspect of the paper. Fundamentally, we do reframe the coarse-grained localization problem as a multi-label classification problem. Furthermore, we are able to use the activations of our neural network to provide even finer-grained information about where the abnormality is located (e.g. Fig 5d-e), even though we do not have any training data with boundaries indicated. In cases where providing such boundaries is extremely labor-intensive, our approach provides useful information (in the style of Saab et al., 2019) about coarse-grained localization via a multi-label formulation, and additionally provides fine-grained information via the neural network activations.

Saab, Khaled, et al. "Doubly Weak Supervision of Deep Learning Models for Head CT." *International Conference on Medical Image Computing and Computer-Assisted Intervention*. Springer, Cham, 2019.

R1.6 Multi-task: the paper does not compare the single-task and multi-task settings using the text-mined labels. Without such, I cannot review the contribution of multi-task learning in this study. It seems that Fig 3b compares single-task and multi-task settings. Unfortunately, Fig 3b is not discussed in the paper.

The original manuscript did attempt to make this comparison between single-task and multi-task models, but it was not as clearly described in either the text or the figures as we had intended. We thank the reviewer for pointing this out, and have updated the manuscript accordingly. Specifically, the new Figure 4 (see R1.8) is focused almost

entirely on this comparison. In Figure 4a, we show this comparison for the twenty-six primary regions in our model and demonstrate how the benefits of multi-task learning are usually higher in regions where abnormalities are rarer. Figure 4b makes the comparison on the task of rare abnormality detection – we find that multi-task learning enables an average improvement of 24.1 AUROC points on four regions for which we have very few positive examples. Additionally, in Figure 4c, we show how multi-task learning greatly reduces the computational cost required to train abnormality localization models. Finally, Figure 4d demonstrates that the representation learned via multi-task learning can have (but does not always have) substantial advantages over a single-task representation for predicting mortality. Note that even the mortality prediction result obtained using the the single-task FDG-PET/CT model represents a novel result to the best of our knowledge. The numbers behind these four figure panels are provided in Supplementary Tables 2, 6, 7 and 8.

R1.7 Weak supervision: The paper compares the proposed models with (1) Kinetics model and (2) summary code model. However, I feel the comparison is not fair. First, although both baseline models are fully-supervised, they are single-task models. Second, neither of them predicts 26 categories. I would like to see a comparison of models training on (1) a small annotated training set and (2) this small annotated set plus large-scaled text-mined training set.

We agree with the reviewer that these comparisons would help readers understand the trade-off between weak supervision and full supervision. To make comparison (1), we first retrained all of our models on a small hand-labeled dataset of 400 exams. We find that models trained on small hand-labeled datasets struggle to detect abnormalities in most parts of the body. The results of this analysis are shown in a new panel, Figure 3b. We've added a paragraph to the results describing these results:

To characterize the trade-off between weak and full supervision, we train the same multi-task abnormality localization model described above on a manually labeled training dataset of 400 exams. This labeling procedure, which requires annotating twenty-six anatomical regions for the presence of hypermetabolic abnormality, took 16 board-certified radiologist hours. In comparison, our weak supervision approach required only 12 non-expert hours to label a dataset of 6,530 exams. Despite the reduced labeling effort, our model trained on weak labels outperforms the model trained on hand labels in every anatomical region (Figure 3b and Supplementary Table 3). On average, weak supervision enables a 22 point increase in AUROC over the fully-supervised model ($p = 0.0000$, paired permutation test). In the liver, for example, the weakly supervised model detects abnormalities with an AUROC of 92.6%(91.8%, 92.4%), while the fully supervised model detects these same abnormalities with an AUROC of 57.4% (52.4%, 61.2%). In nine anatomical regions, including the neck, iliac lymph nodes, and abdomen, the fully-supervised model is no better than random. Indeed, the

increased dataset size enabled by weak supervision leads to considerable performance gains over fully supervised approaches that use the same or more labeling resources.

(See line 212 to line 226)

The reviewer also recommends evaluating a hybrid approach – comparison (2) – that combines weak and full labels. For this evaluation, we retrained all of our models on a dataset combining weakly and fully supervised training examples:

We also evaluate a hybrid approach that combines weak supervision and manual labeling. This model is trained on a mixed dataset consisting of 6,530 weakly labeled exams and 400 hand labeled exams. On average, this hybrid approach enables an improvement of 1 AUROC point over weak supervision alone – a statistically insignificant difference ($p = 0.2719$, two-sided paired permutation test). While we do see performance gains in some regions (e.g. lungs), the boost in performance is usually small.

(See line 227 to line 232)

R1.8 Rare abnormality detection: while the proposed method achieved 77% of AUC, there is no baseline (i.e., a single-task model) to understand the level of difficulty of this problem. Some rare abnormalities may be easy to detect, and a few positive training examples are enough to train the model. To prove the hypothesis that the proposed framework helps overcome the lack of data, it would be important to compare the proposed model with a baseline.

We appreciate the reviewer’s question. The original manuscript did make this comparison to a single task baseline in Figure 3c, but it was poorly labeled in the legend. The original legend included three strata “Pretrain on Kinetics”, “Pretrain on Summary Codes”, and “Pretrain on weak labels (multi-task)”. The first two strata both correspond to single task models that use different pre-training strategies. The last stratum corresponds to multi-task models with task-specific fine-tuning.

In the new manuscript, the figure has been moved to Figure 4b and includes updated strata labels that better communicate the point of the figure. The new strata are “Single Task (Kinetics Pretraining)”, “Single task (in-domain pretraining)” and “Multi-task (with fine-tuning)”, respectively. Figure 4b shows that the single task baselines – even when supervised using the hand-provided summary codes (i.e. single-task, in-domain pretraining) – perform substantially worse than the multi-task model in these four regions in which hypermetabolic abnormality is rare.

R1.9 The organization of the results needs to improve. The authors are recommended to highlight the most important observations in the Results section.

We’ve reorganized the Results section to better highlight our key findings. The new manuscript includes a subheading for each major result: (1) “Combining weakly supervised multi-task learning with spatial attention mechanisms enables automated abnormality detection and localization in FDG-PET/CT”, (2) “Weak supervision reduces

labeling costs for automated abnormality detection and localization”, (3) “Multi-task learning improves automated localization performance, reduces computational cost, and facilitates mortality prediction”, (4) “Spatial attention mechanisms improve automated localization performance and facilitate model interpretation.” We’ve also updated the content and presentation of our analyses to more clearly support these specific statements of our results; for instance, we have updated the figure organization to directly align with each of these sections, and made analogous changes to the text. See responses to R1.7 and R2.6, for examples.

R1.10 The organization of the figures needs to improve. It is hard for me to follow the logic between subfigures. For example, Figure 3 contains five different topics: performance of NLP models, the relation between AUC and training data size/epochs in the image models, rare disease detection, and mortality prediction. I am not sure the whole picture the authors would like to show.

We appreciate the reviewer’s feedback on the clarity of our figures. As described in R1.9, we have changed the organization and flow of our figure panels to better communicate the most important points in our work. Notably, we’ve split up Figure 3 into two new figures. The new **Figure 3** focuses on how our proposed **weak supervision** framework is critical for achieving strong abnormality detection performance: panel (a) shows how language modeling enables better weak label generation, panel (b) shows how abnormality localization models trained with a small, fully-supervised training set perform far worse than those trained with weak supervision over a larger dataset, and panel (c) shows how weak labels can support models that perform similarly to those trained using manual labels across a wide range of dataset sizes. The new **Figure 4** focuses on the importance of **multi-task learning** in our framework. Panels (b) and (d) show how multi-task learning enables performance gains in abnormality localization and mortality prediction. Panel (a) shows how the benefits of multi-task learning tend to increase as the normal/abnormal class imbalance worsens. Panel (c) shows how multi-task learning greatly reduces the computational cost required to train abnormality localization models. Finally, as before, Figure 5 focuses on the effect of our proposed task-specific spatial attention approach and its utility in model interpretability. We have also updated the wording and graphics across all five figures to improve clarity.

R1.11 Fig 3e is before Fig 3a-d

Figure 3e is now Figure 4d in the new manuscript and is no longer mentioned out of order in the text.

R1.12 Fig 3b is not discussed in the paper.

Figure 3b (Figure 3c in the updated manuscript) is discussed in the paragraph spanning line 239 to line 246, however the word “Figure” was missing from the sentence “In Figure 3b, we explore how...”. We thank the reviewer for drawing our attention to this. We have fixed the typo.

R1.13 Fig 3e. DoD is not explained in the paper.

We've updated this axis label to read "date of death" instead of "DoD".

R1.14 Supplementary Figure 1b-d. The max follow-up time is 3500 days. Should it be 365 days?

The latest follow-up in our retrospective data falls 3,500 days after the PET/CT exam, so the maximum value on the x-axis of these Kaplan-Meier curves should indeed be 3,500.

R1.15 It is unclear how the CoxPH model was trained to predict the overall survival probability of mortality.

The updated manuscript includes a new Methods section "Mortality prediction and survival analysis", which should clarify how we fit our Cox models. The section details which covariates were included in the Cox models and a description of the statistical tests we used to evaluate them.

Mortality prediction and survival analysis. To evaluate the capacity of our scan model to predict mortality from PET/CT imaging data alone, we frame mortality prediction as a binary classification task...:
(See line 780 to line 841)

R1.16 "We demonstrate that it is the multi-task formulation, not just in-domain pre-training, that enables these performance gains". The comparison between multi-task on text-mined labels and single-task on summary codes is not fair. There are two variables: learning strategy and training labels. Again, a baseline of text-mined labels in a single-task setting is needed.

We appreciate the reviewer's point and have added this comparison; the multi-task model on text-mined labels outperforms the single-task model on text-mined labels by an average of 15.8% (10.7 points AUROC). See added text (reproduced below) for details.

Additionally, we demonstrate that it is the multi-task formulation, not just in-domain pre-training, that enables these performance gains: compared to models pre-trained ~~on FDG-PET/CT summary codes in a single-task setting (see *Methods*)~~, our multi-task model achieves a 54% mean AUROC improvement (50.5% to 78.0%) in detecting abnormalities in the celiac lymph nodes, a 52% improvement (59.3% to 89.9%) in the kidneys, a 34% improvement (58.2% to 77.9%) in the adrenal glands, and a 18% improvement (66.4% to 78.2%) in the pancreas. using weak labels on the task of whole body abnormality detection, our multi-task model achieves a 15.8% mean AUROC improvement (67.3% to 78.0%) in detecting abnormalities in the celiac lymph nodes, a 25.4% improvement (71.7% to 89.9%)

in the kidneys, a 28.9% improvement (58.2% to 77.9%) in the adrenal glands, and a 26.17% improvement (62.0% to 78.2%) in the pancreas.
(See line 267 to line 841)

R1.17 The authors used k-gram to add the regions to the ontology. The authors didn't explain how the edges were added.

On line 548, we explain of how edges connect regions to sub-regions:

Our ontology is a directed-acyclic graph where nodes represent regions and edges connect regions to sub-regions. For example, edges lead from “lungs” to “left lung” and from “thoracic lymph node” to “hilar lymph node”.

Coauthors G.D. and M.L. specified what constituted a sub-region and G.A. and S.E. encoded these regions as edges in the implementation. We've added a sentence on line 555 in the methods explaining this process:

The edges connecting regions to sub-regions were added in consultation with nuclear medicine and radiology specialists (G.D. and M.L.).

R1.18 The authors of BERT recommend using the original vocabulary to keep fine-tuning the BERT model. This is partly because the initial weights of BERT_base rely on the vocabulary. In this article, the authors used a new vocabulary (with additional 3,000 wordpiece tokens) to fine-tune the BERT model. The authors need to explain the reason, how it will affect the results, and provide the comparison.

The initial BERT weights do indeed rely on the vocabulary, which is why we do not replace the original vocabulary with a new one, but instead add on an additional 1,325 tokens to the original vocabulary using the “[unusedX]” and non-ASCII tokens provided in the original implementation, as described below; this is standard practice when expanding a BERT vocabulary, and allows us to use the initial BERT weights while also introducing domain specific tokens (e.g. hypermetabolic). We've clarified our reasoning in the Methods section:

To account for the specialized FDG-PET/CT text, we use a greedy algorithm to find the set of 3,000 wordpieces that minimize the number of tokens required to reconstruct the reports in our training dataset [36, 38]. Of these 3,000, 1,675 are already in the BERT vocab. We add the remaining 1,325 wordpiece tokens to the original vocabulary by replacing the “unusedX” and non-ASCII tokens provided in the BERT implementation. This allows us to leverage the initial BERT weights while also introducing domain specific tokens. Note, it is important to exclude any testing data when generating word pieces to ensure that evaluations on the test set provide a fair estimate of generalization error.

(See line 601 to line 608)

R1.19 While the authors explain the summary code of 1, they didn't explain the meaning of the codes 2, 4, and 9.

We've added more details on the meaning of these codes in the Methods section:

“The summary code describes overall patient status and takes on a value of 1, 2, 4, or 9, where 1 indicates no evidence of abnormality, 2 indicates that there are abnormalities but they were known prior to the exam, 4 indicates that there is at least one new abnormality not previously known, and 9 indicates the presence of an abnormality that requires emergent attention.”

(See line 490 to line 494)

Furthermore, is it a single-task model or a multi-task model?

It is a single task model. We've clarified that in the methods:

We derive binary abnormality labels from these summary codes and train a single task model to detect abnormalities in the full scan.

and in the results:

We use these summary codes to train a fully-supervised, single-task model for whole-body abnormality detection.

R1.20 The authors created an ontology of 96 anatomical regions, but the multi-task model can only predict 26 anatomical regions. The authors should explain how these 26 out of 96 are chosen. The authors also need to explain if the 96 regions are too refined.

Due to gradient interference between tasks and the severe class imbalance in most regions, we found it challenging to train one multi-task model on 94 different regions (we have corrected a typo from the original manuscript that said 96 instead of 94). Thus, during multi-task training we instead use the 26 regions with the highest prevalence of abnormality. However, this does not mean that our framework can't generalize to more refined regions beyond the main 26 (e.g. celiac lymph node). As we show in our analysis on rare abnormality detection, we can achieve strong performance on regions beyond the main 26 by fine-tuning the multi-task representation on these rarer regions.

We've made several changes to the manuscript to better communicate these details. In the results, when we first mention the 26 anatomical regions, we added details on how these regions were selected on line 144:

Using these probabilistic labels, we train an attention-based, multi-task 3D convolutional neural network (CNN) to detect abnormal metabolic activity in each of the 26 anatomical regions with highest prevalence of abnormality in our dataset.

(See line 144)

Later on in the results, we show how we can train performant models on some of the most refined regions in the ontology by fine-tuning on top of the 26-region multi-task representation:

Our labeling framework generates labels for 94 anatomical regions, many of which are so refined that we lack enough positive examples in our dataset to train performant single-task models. In Fig. 4a, we show that multi-task learning mitigates some of the issues associated with this large class imbalance and enables us to train performant scan models, even on some of the regions for which we have the fewest positive examples. To demonstrate this, we chose four clinically relevant regions beyond the main 26: celiac lymph nodes (68 positive training examples, 84th most out of 94 regions), kidney (182 positives, 64th most), adrenal gland (196 positives, 60th most), and pancreas (201 positives, 58th most), and compare model performance across various types of supervision in Fig. 4b. On all four, our multi-task models substantially outperform single-task models pre-trained on Kinetics. In terms of mean AUROC, we see a 56% improvement (50.2% to 78.0%, $p = 0.0003$ paired permutation test) in detecting abnormalities in the celiac lymph nodes, a 41% improvement (63.5% to 89.9%, $p = 0.0022$ paired permutation test) in the kidneys, a 34% improvement (58.0% to 77.9%, $p = 0.0004$ paired permutation test) in the adrenal glands, and a 40% improvement (55.9% to 78.2%, $p = 0.0000$ paired permutation test) in the pancreas.

(See line 253 to line 281)

R1.21 The original images are 128 x 128, while the input of CNN is 224 x 224. Will upsampling from 128*128 to 224*224 introduce noise to the PET image? What technique did the authors use for the upsampling?

We use bilinear interpolation in order to upsample the PET image for CNN compatibility, and the main body text now clarifies this point. The reviewer's concern is well-founded and we have run further experiments in order to clarify the effect of upsampling. In response to the reviewer's comment, we added an ablation found in the supplementary information that demonstrates that models that forego the upsampling step underperform relative to our models, which use upsampling. In this new experiment, we trained one novel multi-task model on the 26 tasks by downsampling the 512 x 512 CT images and maintaining the PET images to attain a resolution of 128 x 128 for both modalities, then fine-tuned 4 single-task models which predict abnormality on (1) the full body, (2) the liver, (3) the lungs, and (4) the inguinal lymph nodes. We compared these models to our models, which downsample the CT images and upsample the PET images to 224 x 224, and note that our models outperform the models in this new experiment by up to 4 AUROC points. This is shown on line 719:

We randomly crop the image sequence to a 200×200 pixel region, then resize to 224×224 pixels, downsampling the \$512 \times 512\$ CT images and

upsampling the 128×128 PET images using bilinear interpolation. We additionally jitter the brightness of the image sequence by adjusting brightness throughout the sequence by a factor $\gamma \sim \text{Uniform}(0.0, 0.25)$. Further details regarding the training setup can be found in Supplementary Note 3, and ablations that demonstrate the effect of upsampling the FDG-PET images and excluding the CT modality during training can be found in Supplementary Tables 10 and 11, respectively.

(See line 719)

R1.22 “over the last five years there has been a 16-fold increase in number of FDG-PET/CT examinations within the United States”. Please provide a citation.

We thank the reviewer for pointing out this error, which was an unintended holdover from an early draft. We have updated this statistic and provided a citation.

R1.23 Comparing this paper with previous papers that use text-mined labels to classify medical images is necessary. For example, 1. Gong T, et al Automatic labeling and classification of brain CT images. In 2011 18th IEEE International Conference on Image Processing 2011 Sep 11 (pp. 1581-1584). 2. Singh S, et al. Deep-learning-based classification of FDG-PET data for Alzheimer’s disease categories. In 13th International Conference on Medical Information Processing and Analysis 2017 Nov 17 (Vol. 10572, p. 105720J). 3. Yan K, et al. Holistic and comprehensive annotation of clinically significant findings on diverse CT images: learning from radiology reports and label ontology. CVPR 2019 (pp. 8523-8532).

Thank you for these citations. We have included comparisons with these works in our related work section. Additionally, we have added a comparison with Dunmon et al., 2020, which leverages text-mined labels in a similar fashion to our work.

Dunmon, Jared A., et al. “Cross-modal data programming enables rapid medical machine learning.” *Patterns* (2020): 100019.

R1.24 Multi-task learning is widely used in medical image analysis, some related work of multi-task learning on medical image analysis is needed. For example, 1. Khosravan N, Bagci U. Semi-Supervised Multi-Task Learning for Lung Cancer Diagnosis. *Conf Proc IEEE Eng Med Biol Soc.* 2018;2018:710713. doi:10.1109/EMBC.2018.8512294. 2. Moeskops, Pim, et al. “Deep learning for multi-task medical image segmentation in multiple modalities.” *MICCAI 2016.*

We thank the reviewer these citations as well. We have also included these in our related work. However, we want to emphasize a major difference between our work and the references provided by the reviewer. Specifically, our work leverages weak supervision to enable multi-task learning across a large number of tasks (26); the provided references leverage only two or three tasks, a full order of magnitude fewer, and often use expensive manual segmentation labels to supervise those tasks. In this

context, our work represents a move towards the type of “massive multi-task learning” described in Ratner et al., 2019, where scalable supervision approaches such as those presented here enable multi-task learning over numbers of tasks far larger than would be amenable to manual labeling. From a novelty standpoint, we argue that this type of approach has rarely been applied in medical applications, and certainly not in the type of volumetric imaging analysis we propose here. We have also added this citation to the text and remarked on our incorporation of “massive multi-task learning” principles in this work to further differentiate it from previous efforts.

Ratner, Alexander J., Braden Hancock, and Christopher R. ”The Role of Massively Multi-Task and Weak Supervision in Software 2.0.” CIDR. 2019.

Responses to Reviewer #2

R2.1 The manuscript "Multi-task weak supervision enables automated abnormality localization in whole-body FDG-PET/CT" by Eyuboglu et al. provides a generalizable framework for automated detection of abnormalities from FDG-PET/CT scans based on complicated machine learning and natural language processing models. The framework is highly interesting and can prove very useful for generating novel computational tools to help nuclear medicine specialists.

We thank the reviewer for taking the time to understand our contributions and for pointing out that our framework is generalizable to clinical settings beyond PET/CT.

R2.1 The article in itself presents the methodology and results adequately, but the content is very densely written, and therefore may be difficult to understand for some readers. This is not necessarily a limitation. Balancing adequate scientific content with readability is difficult in a complex area like this.

We appreciate this comment, and have made significant changes to the prose throughout the manuscript and reorganized all of the figures and results to improve flow and readability. See responses to R1 for details.

R2.2 The method presented is truly novel and a very compelling alternative to labelling PET voxels for machine learning algorithms.

We thank the reviewer for highlighting the technical importance of our work.

R2.3 Although the presented framework enables training of an automated abnormality localizer based on whole-body FDG-PET/CT scans and weak-supervised learning without the need for manually entering abnormality locations in the training data, future models still need to be properly validated. The authors used a self-labelled test dataset to validate their model, which is a requirement also for future models in other settings. To avoid the perception that no hand labelled data are need for validation purposes, the authors should touch upon this in the discussion.

The reviewer raises an important point that definitely should be addressed in the discussion. We have added the following paragraph.

Second, although training abnormality localization models with our framework does not require manually annotating large amounts of training data, like with all deployed clinical machine learning frameworks, routine validation on a gold-standard, prospective dataset is still critical for detecting degradation in model performance caused by shifting patient populations or changing imaging protocols. Weak labels would not suffice for these evaluations, which means that deploying models in practice will require some

manual labeling of test data. This critical step will likely require significant effort: hand-labeling our test set of 800 exams took 32 radiologist-hours. That being said, if an evaluation indicates that model performance has degraded, our framework makes it easier to integrate new training data to address the issue. Note that these evaluations would be required for any automated interpretation system that would be deployed in clinical practice, and thus the requirement is not specific to the models we propose.

(See line 449 to line 459)

R2.4 It is not entirely clear what role the validation dataset plays in the framework. Presumably, it used for the so-called fine-tuning step, but this is not explicitly mentioned. The authors should elaborate.

We've added a few sentences to the Methods clarifying the use of the validation set (first on line 522, second on line 716:

The validation set was used to evaluate prototypes during model development, to tune hyperparameters, and for early stopping during model training. All reported metrics were computed on the test set, unless otherwise specified.

After fine-tuning, we use the model weights from the epoch with the highest validation AUROC.

The models are fine-tuned on training data, not validation data. In the Methods, we clarify how the validation set is involved in fine-tuning:

Through this fine-tuning step, we are able to (a) improve performance in the main 26 regions and (b) train performant models on regions beyond the 26 most prevalent (see Results). We perform fine-tuning for 5 epochs. After each epoch, we evaluate on the validation set using weak labels.

(See line 449)

R2.5 The authors could improve the readability of the results, if they were more explicit about which dataset and endpoint are used when reporting AUROC, sensitivity, F1, etc. throughout the manuscript. For instance, in the sentence "Our labeling framework enables a 28% improvement in mean F1 score (52.2% to 66.6%) over this regular expression baseline and achieves a mean AUROC of 90.0%" on page 8, it is not clear whether this is based on validation data or test data, or whether this is based on the downstream scan output or the upstream labels.

To clarify, we've added the following sentence under the heading "Model evaluation" in Methods.

Unless otherwise specified, all reported metrics in results were computed using data from the test set and the hand labels described above. No hyperparameter tuning or model development was performed with the test set. (See line 524)

The sentence the reviewer pointed out might be particularly confusing, since throughout most of the paper we are evaluating the scan model, except in that single section where we evaluate the report model. So, we now clarify which data and labels we use when evaluating the report model.

We observe a mean AUROC of 90.0% when evaluating weak labels emitted by our labeling framework against hand labels extracted from the same reports by radiologists (see Methods); further, we find a 28% improvement in mean F1 score (52.2% to 66.6%) with respect to a regular expression baseline. With a threshold of 0.5, our model achieves a mean sensitivity of 87.0% and a mean specificity of 82.2%, outperforming the regular expression baseline, which achieved a sensitivity of 83.3% and a specificity of 66.9%. Note that we compare our labeler to the regular expression baseline using F1-score (not AUROC) because the baseline doesn't output scores, just binary predictions.

In fact, it would be more interesting to look at the downstream scan output, as the presented framework does not suggest to use the labeler alone.

We agree with this point, and have clarified our presentation. Most sections in the paper are indeed focused on evaluating the performance of the downstream FDG-PET/CT image model (i.e. not the language model applied to the report text). Our evaluation in this section is meant to demonstrate that our labeling framework can accurately extract what the radiologist documented in the report, independent of what is in the scan. I.e. we do not argue that our language model should ever “correct” what was in the report – rather, it should simply extract what a knowledgeable radiologist already recorded. We've added a sentence to the beginning of the section clarifying the purpose of this particular exploration:

First, to evaluate the capacity of our labeling framework to accurately extract abnormality locations from radiology reports, we compared the labels generated by our framework to the hand-labels manually extracted from the reports by radiologists.

R2.6 Although a secondary endpoint, the authors propose to predict x-day mortality based on the presented framework. However, there is no mentioning in the manuscript of how patients were followed and how the authors handled censoring (if any).

The original manuscript lacked sufficient details on the mortality data used in this study. We added a section in the methods describing how we acquired dates of death for the patients in our cohort and handled censoring

This study was approved by the Stanford Institutional Review Board. The Stanford Research Repository (STARR, formerly STRIDE) is Stanford Medicine’s approved resource for working with clinical data for research purposes [29]. FDG-PET/CT imaging data and radiology reports for patients in the cohort were acquired with STARR and deidentified. Additionally, dates of death for a subset of the cohort were retrieved using STARR, which integrates medical records at Stanford with the Social Security Death Index (SSDI). Of the 4,691 patients in our dataset, 867 (18%) had a date of death recorded in STARR. Patients were not prospectively followed for clinical outcomes in this retrospective study design. Patients without a date of death recorded in STARR were right censored at the time of their last PET/CT exam in our data.

We also added a few sentences to the discussion highlighting the limitations of using censored, retrospective data for mortality prediction.

Third, in this study, we show how our scan model can be fine-tuned to predict mortality from FDG-PET/CT scans. We chose to train mortality prediction models on imaging data alone not because doing so would necessarily make sense in clinical practice, but because it demonstrates the utility of this data source in mortality prediction. In practice, palliative care decisions should be based on a number of different factors, but our work shows for the first time that automated analysis of FDG-PET/CT data could be useful in this setting. Patients were not prospectively followed for clinical outcomes in this retrospective study design. Dates of death were acquired retrospectively and patients without a date of death were censored in our analyses. Future studies should evaluate mortality prediction models that incorporate learned representations of FDG-PET/CT data in a controlled clinical trial population with specific follow-up protocols designed for the purposes of this research question.

The modelling approach in this setting explained in the sentence, “Our mortality prediction model is pre-trained on multi-task abnormality detection in 26 anatomical regions and fine-tuned on a small dataset that pairs each study with a date of death” is challenging to grasp. Does “study” mean location? What does it mean to pair with date of death?

A “study” is a single PET/CT exam. By “pairing with a date of death” we mean associating the PET/CT exam with the patient’s date of death so we can use the exam as a mortality prediction training example. We’ve made the following changes to the explanation:

We frame mortality prediction as a binary classification task, where the model predicts whether the patient’s date of death is within x days of the study given only the FDG-PET/CT scan as input. In our experiments, we set

x to be 45, 90, 180, and 365 days. Our mortality prediction model is first pre-trained on multi-task abnormality detection in 26 anatomical regions ~~and fine-tuned on a small dataset that pairs each study with a date of death” is challenging to grasp~~. Then, the model is fine-tuned to predict mortality within x days using the subset of PET/CT exams in our data for which we have a recorded date of death. We fine-tune a separate model for each time point.

And does the same model predict mortality at all four time points?

We train a separate model for each time point. We’ve added a sentence to make this clear:

We frame mortality prediction as a binary classification task, where the model predicts whether the patient’s date of death is within x days of the study given only the FDG-PET/CT scan as input. In our experiments, we set x to be 45, 90, 180, and 365 days. Our mortality prediction model is first pre-trained on multi-task abnormality detection in 26 anatomical regions ~~and fine-tuned on a small dataset that pairs each study with a date of death” is challenging to grasp~~. Then, the model is fine-tuned to predict mortality within x days using the subset of PET/CT exams in our data for which we have a recorded date of death. We fine-tune a separate model for each time point.

As a baseline, the authors could also compare with a model, that only utilizes the patients age, as this is much easier to obtain than FDG-PET/CT scans. If age is as good a predictor, there might be limited utility in basing mortality predictions on scans.

The reviewer’s point, that simpler covariates like age might be as good a predictor of mortality as the locations of abnormalities, is a good one. To explore this issue, we fit a Cox proportional hazards model on age and a few other simple covariates and compared it to a Cox model that uses FDG-PET/CT abnormality localization predictions as covariates. See response to R2.8 for more details.

R2.7 While impressive mortality prediction, one could be concerned about using imaging alone for this purpose. Imaging does not provide the diagnosis and while the distribution of lesions may hint to final diagnosis, relying on this for palliative care decisions would be too uncertain and could risk patients with curable disease such as lymphoma to be assigned to palliative care. We are aware that this is not proposed for direct use by the authors, but more limitations are needed for this part.

We agree with the reviewer: we need to emphasize that we chose to train mortality prediction models on FDG-PET/CT data alone not because doing so would make sense in clinical practice, but because it demonstrates the utility of this unconventional data source in mortality prediction. We’ve added a few sentences to the discussion on this topic:

Third, in this study, we show how our scan model can be fine-tuned to predict mortality from FDG-PET/CT scans. We chose to train mortality prediction models on imaging data alone not because doing so would necessarily make sense in clinical practice, but because it demonstrates the utility of this data source in mortality prediction. In practice, palliative care decisions should be based on a number of different factors, but our work shows for the first time that automated analysis of PET/CT data could be useful in this setting. Patients were not prospectively followed for clinical outcomes in this retrospective study design. Dates of death were acquired retrospectively and patients without a date of death were censored in our analyses. Future studies should evaluate mortality prediction models that incorporate learned representations of FDG-PET/CT data in a controlled clinical trial population with specific follow-up protocols designed for the purposes of this research question.

(See line 460 to line 470)

R2.8 The authors state that: "The importance of abnormality localization pre-training in mortality prediction could be explained by the observation that patients for which the model detected at least one abnormality showed a significantly lower survival". However, this does not explicitly suggest a benefit of detection the locations of abnormality, as this is not what the authors investigate with the Cox model. Also, the fact that lesions detected associates with mortality is not so surprising. Likely, just the fact that having a PET done is predictive of death in broad population, among those with PET done – abnormalities are predictive of death etc. I would be good to include more on this part, what type of abnormalities are strongly associated with death – and when adjusting for baseline diagnosis - is the automated PET still important?

Reviewer 2 raises several interesting questions regarding the importance of abnormality localization when other covariates, such as age or disease, are more readily available. We have expanded our survival analysis to shed light on these questions. A results section describing our findings was added:

To understand why multi-task abnormality localization pretraining improves mortality prediction performance, we perform a survival analysis that explores the relationship between abnormality localization predictions and longevity. We fit a Cox proportional hazard model on our mortality data using PET/CT abnormality localization predictions as covariates (see *Methods* for details, Supplementary Table 18 for hazard ratios). Via the likelihood ratio test, we confirm that the fit is significantly better than a null model with just a baseline hazard and no covariates ($p = 1.1 \times 10^{-12}$, likelihood ratio test). The covariates corresponding to hypermetabolic abnormalities in the liver, spine, skeleton, and the hilar, inguinal, and thoracic lymph nodes exhibit statistically significant hazard ratios (Wald test). In Supplementary Figures 1c-d, we show Kaplan-Meier curves stratified by

the predictions for the liver and hilar lymph nodes.

The locations of hypermetabolic abnormalities appear to be predictive of mortality, but is this information still useful when we have access to other more readily available covariates like age or disease type? To explore this question, we fit a Cox proportional hazards model on our mortality data using abnormality localization predictions, age, indication, and exam summary codes as covariates (see *Methods* for details, Supplementary Table 16 for hazard ratios). We also fit a nested Cox model that includes all covariates except the abnormality localization predictions (see Supplementary Table 17 for hazard ratios). Via the likelihood ratio test, we show that the fit with abnormality predictions is significantly better than the fit without ($p = 3.7 \times 10^{-11}$) [22]. We also compare the out-of-sample predictive power of these Cox models by fitting on the validation set and computing the concordance index on the test set. A Cox model fit just on age, summary code and indication achieves a concordance index of 0.609, whereas a model that incorporates abnormality location prediction achieves a concordance index of 0.720. Indeed, the locations of hypermetabolic abnormalities seem to provide useful signal not present in other, more simply-attained covariates such as age or indication.

(See line 309 to line 337)

as well as a detailed description of our approach in the methods:

We also perform a more traditional survival analysis to explore whether the abnormality localization predictions are predictive of mortality. We fit Cox proportional hazard models on our mortality data using abnormality localization predictions, age, indication, and exam summary code as covariates...

(See line 790 to line 841)

R2.9 The authors calculate the performance metrics over five random seeds (leading to five different models?), as the training procedure has some stochastic elements to it. The authors could reflect a little on how this should be handled in everyday use. Should many models be generated and the mean (over each model) of predicted probabilities be used as the final prediction? The authors could discuss this.

We have expanded our discussion to include the performance of an ensemble model over the five random seeds, which shows a performance gain on each task of up to 3 AUC points. See line 444.

R2.10 The reliance of imaging reports for modelling is extremely interesting, but it is important to know more about the report structure. Are they following the same structure or are reports more in free text? In some countries, a report would be free text, others more checklist like. I can imagine the latter would provide better foundation for a work like the present. Could the authors show the structure of a report in the paper? Is it a fixed format?

We appreciate the reviewer’s enthusiasm for our method, and have attempted to provide more information here.

“Are they following the same structure or are reports more in free text?” The reports are free text with a relatively consistent set of subsections. The subsections include the “Clinical History,” “Patient Demographics,” the radiologist’s “Findings,” and a final summary section known as the “Impression.” – see line 503.

“Is it a fixed format?” We exploit the consistency of these subsections to extract the radiologist’s findings and final impression in order to train our natural language model.

R2.11 Can we have more information on the patients that are examined? Cancer patients? Which types. It would be particularly useful to describe if the model can predict type of cancer and whether it is localized or not? In some cancers, having lower sensitivity for some areas, for example for mediastinal lymph nodes may matter a lot, for example in lung cancer. How does the model perform for such tasks, can this be assessed? Helping physicians where to guide biopsy in case of predicting cancer type or as an initial assessment of cancer stage (localized vs non-localized)

“Can we have more information on the patients that are examined?” Due to dataset de-identification that occurred even before we received it, we are limited in our ability to elaborate upon the patient characteristics in this manuscript. However, in response to this comment, we do provide the following: (1) patient age, (2) the study date, (3) various machine characteristics, and (4) the cancer type derived from the Study Description entry in the DICOM headers of each exam. The exam breakdown of this information for the training, validation, and test datasets can now be found in Supplementary Tables 12-15. Further, we have updated Supplementary Table 1 to show the abnormality prevalence for each region as annotated for the training, validation, and test sets. These supplementary materials are presented in the main body of the essay on line 494 and line 667.

“It would be particularly useful to describe if the model can predict type of cancer and whether it is localized or not?” We appreciate the reviewer’s interest in potential use cases for our research. While we have incorporated key patient demographic information to the supplement, including cancer type, in response to the reviewer’s comment, we do not have the primary tumor location for all elements of the training set as structured data and acquiring such information would be a major effort. However, we recognize that this would be a high impact application that would be a great subject for follow-on work.

R2.12 Could the authors elaborate on whether this model will be good for metabolic volume assessments etc. Now that abnormal areas are detected could volumes be determined? It this next step?

“Now that abnormal areas are detected could volumes be determined?” We appreciate the reviewer’s insightful suggestion. We believe that, with a given sample’s saliency

map, it may well be possible to estimate metabolic volume. Indeed, we believe that the ability to create saliency maps for PET/CT scans is a potentially high impact benefit of our approach. Automated metabolic volume assessment is beyond the scope of the paper, but could be excellent follow-on work.

R2.13 Are the generated word-pieces (vocabulary) only based on training data or the entire dataset? They should likely be derived from the training data to ensure fair validation.

Our word-pieces were generated using only the training data. We've updated the methods to emphasize this important point. Also see R1.18 for further details.

To account for the specialized FDG-PET/CT text, we use a greedy algorithm to find the set of 3,000 wordpieces that minimize the number of tokens required to reconstruct the reports in our training dataset [36, 38]. Of these 3,000, 1,675 are already in the BERT vocab. We add the remaining 1,325 wordpiece tokens to the original vocabulary by replacing the “unusedX” and non-ASCII tokens provided in the BERT implementation. This allows us to leverage the initial BERT weights while also introducing domain specific tokens. Note, it is important to exclude any testing data when generating word pieces to ensure that evaluations on the test set provide a fair estimate of generalization error.

R2.14 The authors state that “a baseline labeler that uses this approach achieves a sensitivity of 83.3%, but a specificity of only 66.9%”. Is this in validation/test data or training dataset?

“Is this in validation/test data or training dataset?” The numbers for the baseline labeler are produced using the hand labeled test set. More generally, all results in the manuscript are produced using the held-out test set unless otherwise indicated.

R2.15 Do the authors have an explanation as to why a the fully supervised vision performs slightly worse than the multi-task model. Do the signal get blurred when combining all anatomical regions?

“Does the signal get blurred when combining all anatomical regions?” The reviewer’s intuition regarding the signal in a fully supervised single-task model is correct, in a coarse sense. We hypothesize that the multi-task model outperforms the fully supervised model because (1) it is able to dedicate specific parameters to each region of interest coupled with (2) more fine-grained supervision over each region. The multi-task model architecture incorporates a spatial attention mechanism into each task head, which allows each task head to “focus” on specific regions. Further, the multi-task model is given the labels for each subregion that comprises the full body. These two factors combined lead to more useful gradient updates, which in turn lead to a better representation for full body abnormality prediction.

R2.16 The authors use the term "pre-trained" throughout the manuscript. It might be standard vocabulary in machine learning communities, but in this context it suggests that the model has been trained previously elsewhere and not a part of this study. The authors could make the framework more understandable if they simply wrote "trained".

We thank the reviewer for highlighting the potential for confusion surrounding the term "pre-trained". We do think that it is important to use this term given that it is standard in the machine learning literature and accurately describes the techniques employed by our framework. That being said, we now clarify what we mean by "pre-training" when we first use the term in the introduction on line 103:

We further explore whether multi-task abnormality localization pretraining (i.e. training a model in a multi-task abnormality localization setting before fine-tuning on a different task) can facilitate the development of models for other clinical tasks.

R2.17 How are the sensitivity curves of the perfect detector calculated in Figure 2b-d. The authors could add some information to the figure label.

The figure caption now better explains what we mean by a "perfect detector":

The sensitivity curve of a perfect detector (i.e. a model that ranks all abnormal examples above all normal examples) is shown in grey.

Responses to Reviewer #3

R3.1 While this study has its strengths on the technical/methodological side, it lacks profound clinical considerations. All the challenges attributed to PET/CT scans in the introduction are even more true for CT- or MRI-only examinations. The PET signal is a helpful guidance for abnormality detection, in particular for small and rare findings. This is one of the main purposes to perform a PET/CT.

Our model indeed predicts abnormalities that manifest as hypermetabolic lesions, which are often identified using the PET modality. However, we maintain that our model is an FDG-PET/CT model because our model leverages both the PET and CT modalities simultaneously – as different channels in the input data to the machine learning model – in order to make its final predictions. We have clarified this point in the main body.

However, a PET/CT scan consists of both modalities - each providing unique and often complementary value. Interpreting them individually is not sufficient - thus - also relying on only one of them for anomaly detection is not sufficient because critical and potentially life-threatening information (e.g. pulmonary embolism) can be missed. Therefore, the authors should not refer to the framework as a FDG-PET/CT model but rather a FDG-PET model

The reviewer’s thoughtful insight into the dual modality nature of PET/CT exams inspired a new ablation now included in the supplementary information. We demonstrate that models trained on the PET modality alone in fact under perform relative to the PET/CT models we present, which simultaneously leverage both the PET and CT modalities. In this new experiment, we trained one novel multi-task model on the 26 tasks using only the PET modality, then fine-tuned 4 single-task models which predict abnormality on (1) the full body, (2) the liver, (3) the lungs, and (4) the inguinal lymph nodes. We compared these models to those presented in this paper, which leverage both the PET and CT modalities, and note that our models outperform the models without the CT modality (i.e. FDG-PET only) by 3 AUROC points in the liver ($p = 0.0001$, paired permutation test) and 2 AUROC points in the lungs ($p = 0.0000$, paired permutation test). We update the main body to reference these results on line 723.

R3.2 Overall, I see this study as a proof of concept investigating the possibility to generate automatically labeled training data and perform anomaly detection in PET-scans. The clinical claims such as developing an “effective triage tool” or predicting mortality as “clinically relevant task” to “identify patients that could benefit from palliative care” are exaggerated and simply not supported by the data and analyses.

We appreciate the reviewer’s comment. Our objective with this study is not to develop models in their final clinical form, but rather to introduce a new way of training abnormality localization models that could provide clinical value in a future iteration. For

example, in Figure 3a, we show the importance of using expressive language models as labelers, rather than the rule-based approaches that are commonly employed; in Figure 4, we show how multi-task learning enables strong performance on challenging clinical tasks. Ultimately, because several of our main contributions are methodological and because we do not evaluate the model prospectively, we agree with the reviewer's characterization of our work as a proof of concept. It is our hope that the framework we introduce could empower clinical machine learning practitioners to deploy models in settings where labeling costs would otherwise be prohibitive and/or labeled data for rare pathologies is scarce. However, there remain several important steps needed before deploying and we've highlighted these in the limitations section of the discussion:

While our results are promising in several respects, it is important to emphasize that delivering production-ready models in their final clinical form is beyond the scope of this study, and that there are several additional steps that must be taken before deploying such models in clinical practice...

(See line 428 to line 470)

While we would push back on the general idea that our clinical claims are exaggerated, we take the reviewer's comment as an indication that specific language within our manuscript should be updated to better emphasize that our models are not "production ready". For example, we've made the following changes to our treatment of clinical applications in the introduction:

Our framework enables anatomically-resolved abnormality detection in FDG-PET/CT scans, and ~~provides clinical value in four ways~~ and models based on our framework could potentially provide clinical value in four ways: (1) by automatically screening negatives and enabling the prioritization of exams, (2) ~~by enabling future diagnostic aids that could help radiologists localize abnormalities within the scan, flag rare abnormalities and quickly draft reports.~~ ~~by aiding radiologists in localizing abnormalities within the scan, which is important for quickly and correctly composing reports,~~ (3) ~~by allowing clinicians to quickly identify rare pathology~~, and (3) by yielding a learned representation of each scan that can enable more accurate mortality predictions, which is important for palliative care planning.

(See line 56 to line 64)

Below we walk through each of our clinical claims and specify any changes we've made to clarify our messaging:

(1) *Worklist prioritization/triage*: We discuss in the results and show in Figures 2b-d that models based on our framework could be used for worklist prioritization/triage in a clinical setting. We recognize that the following sentence from the results section in the original manuscript

“One of the primary purposes of this study is to build an effective triage tool for workflow prioritization in PET- CT screening”

may have given the impression that our model is ready to be deployed for this triage use case. We’ve updated this sentence to read:

“Our weakly supervised abnormality localization framework could be used to develop triage tools for workflow prioritization in PET- CT screening.”
(See line 165 to line 169)

Other than this change, we think that the existing language in manuscript provides a fair and objective assessment of our model’s triage capabilities. For example, the rest of the results section on triage reads:

When we sort the predictions by decreasing predicted abnormality, if we only read the first three-quarters of exams, we can still achieve a sensitivity of 95%, 98%, and 96% in the chest, the liver, and the inguinal lymph nodes, respectively. In contrast, when reading exams in the order that they are received we expect to achieve a sensitivity of 75% if we interpret only the first three-quarters of the work-list. This analysis helps contextualize the potential utility of worklist triage in clinical interpretation.
(See line 170 to line 176)

(2) *Mortality prediction*: As we discuss in R2.7, our exploration of mortality prediction was not intended to deliver a “production ready” mortality prediction model. Rather, we aim to show that via weak supervision and multi-task learning unconventional data modalities like PET/CT could be integrated into mortality prediction pipelines alongside more conventional covariates like diagnosis and age (and we include such covariates in our updated manuscript). We’ve added a section to the discussion highlighting the limitations of this aspect of our study:

Third, in this study, we show how our scan model can be fine-tuned to predict mortality from FDG-PET/CT scans. We chose to train mortality prediction models on imaging data alone not because doing so would necessarily make sense in clinical practice, but because it demonstrates the utility of this data source in mortality prediction. In practice, palliative care decisions should be based on a number of different factors, but our work shows for the first time that automated analysis of PET/CT data could be useful in this setting...
(See line 460 to line 470)

(3) *Aiding radiologists localize both rare and common abnormalities within the scan*
The final clinical claim we make is that models based on our framework could enable diagnostic aids that (1) flag rare abnormalities and (2) help radiologists quickly and correctly compose reports. The original language here perhaps suggested that we had

already validated/deployed the model in this use case, when we intended to simply highlight it as a potential use case for this kind of model. We've made the following changes to clarify:

Our framework enables anatomically-resolved abnormality detection in FDG-PET/CT scans, and models based on our framework could potentially provide clinical value in four ways:... (2) by enabling future diagnostic aids that could help radiologists localize abnormalities within the scan, flag rare abnormalities and quickly draft reports. ~~by aiding radiologists in localizing abnormalities within the scan, which is important for quickly and correctly composing reports.~~

(See line 56 to line 64)

If there are other specific clinical claims that the reviewer thinks warrant attention, we would be happy to address them.

R3.3 The authors investigate an interesting concept that could be helpful to address the issue for generating labeled datasets for AI applications in a more efficient way. Overall, the manuscript is well written and easy to read, however, the structure and built up of the different section could be improved. Introduction is too long - especially the last 1 pages should be substantially shortened. Most of it belongs to results or discussion.

We thank the reviewer for giving us feedback on the structure of the paper. We have revised the structure of the introduction, the results, and the figures in an effort to improve clarity throughout. However, given that we've been asked to give more information – not less – by the reviewers in most cases, we respectfully request the reviewer's understanding that we have not been able to substantially shorten the introduction.

Results could benefit from more subheadings.

The results section in the original manuscript included the following subheadings: (1) "Modeling Approach", (2) "Multi-task Weak Supervision Yields Clinically Impactful Models for FDG-PET/CT Analysis" (3) "Multi-task Weak Supervision Formulation Improves Performance, Enables Localization, and Reduces Cost". We've updated the organization of the results, which now follows the figures and includes the following subheadings: (1) "Combining weakly supervised multi-task learning with spatial attention mechanisms enables automated abnormality detection and localization in FDG-PET/CT", (2) "Weak supervision reduces labeling costs for automated abnormality detection and localization", (3) "Multi-task learning improves automated localization performance, reduces computational cost, and facilitates mortality prediction", (4) "Spatial attention mechanisms improve automated localization performance and facilitate model interpretation"

Also, limitations should be summarized in a separate paragraph in the discussion.

We've expanded the limitations in the discussion and moved them to their own paragraphs. We emphasize that there are several additional steps beyond the scope of this study that must be taken before delivering production-ready models. (*See line 428 to line 470*)

R3.4 what about abnormalities in organs/anatomical regions with a high physiological FDG-Uptake? Did the authors consider any threshold for SUV measures to distinguish between abnormal, borderline and physiological findings? For example, in the report excerpt provided in Figure 1a multiple nodes are reported but only a single, moderate SUV uptake is mentioned.

We do not decide on thresholds for abnormality in this study – we implicitly leverage the interpretations of the subspecialist radiologist at the time when the exam was taken, as indications of abnormality in the text report are based on such analysis. We view this to be a strength in our paper– in our analysis, we demonstrate that the multi-task model is able to achieve reasonable performance in abnormality prediction for each region. We posit that the multi-task model internally learns a relationship between FDG uptake and abnormality for each region despite the wide range of mean physiological FDG uptake across regions.

R3.5 Methods section describing setup of Figure 1 should follow labeling order of Figure 1, i.e. 1d follows 1b and 1c.

We've updated the methods section to include references to Figure 1 where appropriate. We also confirmed that the ordering of the methods follows the labelling of Figure 1.

R3.6 Methods: include more information (number of patients and scans) for the training, tuning and testing datasets.

We have added the number of patients and scans for the various partitions of the datasets to the Methods section:

For experiments comparing supervision strategies (see Figure 3b and Supplementary Table 3), we use a hand labeled test set of 423 exams from 235 patients, a validation set of 800 exams from 469 patients, and a training set of 6921 exams from 3987 patients, of which 391 exams from 235 patients are hand labeled. For all other experiments, we use a validation set of 800 exams from 469 patients, a hand-labeled test set of 814 exams from 470 patients, and a training set of 6,530 exams from 3,752 patients.

(*See line 524 to line 529*)

R3.7 Methods, Model evaluation: Did each of the radiologists label all 800 cases of the test set or were they split between radiologists? What was the radiologists experience in interpreting PET/CT scans and how were the labels assigned (probability between 0-1, was there a

training session before actual labelling about how specific expressions should be handled, e.g. “unlikely”, “cannot be ruled out”, “is possible”? Where labels only based on reading the report? How were cases handled if wording was ambiguous?) This is a crucial step that should be described in more detail as the authors themselves mention correctly that the language in PET/CT scans is “quite nuanced”.

We agree that labeling validation and test data is a crucial step in the development of weakly-supervised models and thank the reviewer for pointing out that the original manuscript lacked sufficient details on this topic. We’ve expanded the Methods section entitled “Model Evaluation” to include more detail on the labeling procedure:

Our test set of 800 exams was hand-labeled by four board-certified radiologists (G.D., B.P., A.P., M.L.) with experience ranging from 4 to 15 years. The test set was split among the radiologists such that each radiologist labeled 200 exams. The radiologists labeled each exam based only on the contents of its associated report (i.e. the actual FDG-PET/CT exam was not reinterpreted). For each of 30 anatomical regions (twenty-six main regions plus four rare regions we examine in detail in this work), the radiologists assigned a binary (i.e. $\{0, 1\}$) abnormality label. A label of 1 was assigned to a given region when there was an explicit mention of abnormal FDG uptake in that region in the *findings* or *impression* of the report. For example...
(See line 726 to line 742)

R3.8 No information of patient characteristics are provided in the entire manuscript. This is important information from a methodological as well as a clinical perspective and a major limitation of the study.

Due to the de-identification steps taken on the dataset prior to the start of our study, we are limited in our ability to elaborate upon the patient characteristics in this manuscript. However, we do include what we do have, which includes patient age, the study date, and various machine characteristics. We have provided a breakdown of this patient information in Supplementary Tables 12-15. We are working on recovering additional patient characteristics, and will include as many as possible in the final manuscript.

R3.9 I am missing a separate section summarizing the statistical analyses conducted. The authors mention to use cox models to estimate survival. However, this is only mentioned once in the introduction and once in Figure S1 without providing hazard ratios or any other information about model development and other variables used.

We’ve expanded our discussion of statistical analyses in the Methods. Under the heading “Model evaluation” we’ve added a paragraph describing the statistical tests we employ when comparing deep learning model performance:

Every model in our analysis was trained with five different random initializations (i.e. random seeds). We evaluate each of the five trained models on

the test set and report the mean of the five resulting scores (e.g. AUROC). We account for uncertainty in this estimate of the mean score with 95% confidence intervals computed via bootstrapping over the sample of five scores. When comparing our models to baselines (e.g. weakly-supervised vs. fully-supervised), we test the null hypothesis that the sample of scores of our model comes from a distribution with the same or lower mean than the sample of scores of the other. This is done via a one-sided paired permutation test with $n = 10,000$ iterations. Note when testing whether two models are statistically equivalent, we use a two-sided paired permutation test.

(See line 746 to line 754)

Under the heading “Mortality prediction and survival analysis” we’ve added several paragraphs with details on our survival analysis. There we discuss how we fit our Cox models, which covariates were used and where they came from, and the statistical tests we use to compare covariates and models. Additionally, in Supplementary Tables 16-18 we provide hazard ratios for all models with Wald 95% confidence intervals and p -values.

We also perform a more traditional survival analysis to explore whether the abnormality localization predictions are predictive of mortality. We fit Cox proportional hazard models on our mortality data using abnormality localization predictions, age, indication, and exam summary code as covariates. Unlike the binary mortality prediction experiments described above, patients without a date of death were included in the survival analysis and right censored at the time of their last PET/CT exam in our data...

(See line 790 to line 841)

R3.10 Figure 1: For clarity - Figure 1b should follow the example given in Figure 1a.

We agree with the reviewer that this could be confusing since we are giving an example of a PET/CT report that talks about lymph nodes and the example of abnormality detection using BERT is on a different tissue, lung. We’ve updated Figure 1a to match Figures 1b-c.

I also wonder whether the wording in 1c actually qualifies for a metabolic abnormality. Again, there is no information about the actual SUV uptake.

We understand the concern of the reviewer. However, we do not think that the SUV value is relevant in this case. While historically we have used and continue to use SUV values as an aid to discern benign versus malignant FDG uptake, there are flaws to this method. Sometimes malignancy has low FDG uptake (i.e. low SUV value) and sometimes benign processes have high FDG uptake (i.e. high SUV value). Most of lung parenchyma (tissue) is air. Consequently, typically normal lung tissue has extremely low FDG uptake and SUV values, because there are few cells using the

radioactive glucose (FDG). So FDG-PET images of the normal lungs are photopenic (usually white), as no significant signal appears there. The fact that there is a lung nodule noted on CT (lingular pulmonary nodule) that has increased in size and shows mild FDG uptake (normal lung tissue shouldn't have significant PET signal) renders this an abnormal metabolic finding and warrants follow-up to ensure resolution. This is because mild FDG uptake in a growing lung nodule may be a presentation of low-grade malignancy such as adenocarcinoma in situ.

R3.11 Figure 2: I cannot see the sensitivity curves for the initializations in “light blue”.

The sensitivity curves for the chest, liver and inguinal lymph nodes are actually green, pink and yellow, respectively. The caption has been updated to reflect this and the thickness of the lines has been increased to improve visibility.

R3.12 Figure 4: panel d) if I am not mistaken the described lymph node is in level II on the left not level I on the right.

We thank the reviewer for taking the time to study these FDG-PET/CT images. First, we note that the old Figure 4d is now Figure 5e. The abnormal uptake in the neck (asymmetric) is on the right side and, as the reviewer points out, is likely a level II lymph node given the level of the mandible, this likely is a jugulodigastric lymph node at the right level IIa in the neck. We've updated the caption accordingly.

I also wonder whether the lymph node described in panel e) is axillary or (inter-)pectoral. Please check and revise accordingly.

The old Figure 4e is now Figure 5g. There is indeed an abnormal left axillary lymph node in this study, however it is not seen clearly in the image slice we originally included. We had chosen that image slice because it was the slice with the highest saliency, perhaps due to the artifact from the pacemaker. In response to the reviewer's comment, we've changed the image slice to the one where the abnormality is most clearly visible.

R3.13 Figure S1b: caption does not match order of the panels.

We've updated the figure caption to match the order of the panels.

Could be interesting to add and compare a curve for the summary codes of the original reports in the full body plot.

We thank the reviewer for the suggestion – we agree that this is an interesting comparison. We've added in Kaplan-Meier curves stratified by summary code alongside the curves for full-body abnormality prediction. See Supplementary Figure 1b.

R3.14 Figure S2b: the assigned mean attenuation scores are missing.

We've added the self-attention scores under each token as percentages of the total attention distribution.

R3.15 Correction of typos needed: e.g. Introduction, second paragraph; “FDG-PET demonstrates demonstrates...” or Methods first sentence “FDG FDG-PET/CT...”

Thank you. The changes have been made in the main body text.

Reviewers' Comments:

Reviewer #1:

Remarks to the Author:

Thanks to the authors for kindly attempting to address some of my concerns. The writing and clarity have improved vastly, but I am still concerned about this paper's novelty and the performance of the framework. In addition, I am not sure which concept(s) is (are) proved when the authors claim this work is a proof-of-concept:

- (i) The concept of using NLP to generate weakly labels and then training a DL model for medical image analysis is not new;
- (ii) The concept that multi-task learning outperforms single-task learning has been extensively proved; and
- (iii) the authors did not solve an abnormality "localization" problem (see below).

1. The authors argue that there is no great deal of practical difference between an AUC of 0.8 vs. 0.75, because in this study, "very high sensitivity is required, but nearly any level of specificity would be helpful." This is not true. Both sensitivity and specificity are equally important; otherwise, the authors cannot rank positive examples more highly than negative examples in Fig 2b-d. In other words, the authors provided an analysis of the models' AUC (not only sensitivity) in Figure 2b-d. Please refer to <https://developers.google.com/machine-learning/crash-course/classification/roc-and-auc> for the interpretation of AUC.

2. The authors should not refer to the problem as abnormality "localization". While the framework provides fine-grained labels indicating the localization of abnormality, it does not try to solve an object localization problem. As I have pointed out in my previous review, to localize abnormality in PET/CT, the framework needs to generate pixel-level bounding boxes of abnormal findings. The authors should conduct evaluations by comparing generated boxes with gold references and report standard metrics such as intersection over union ratio (IoU) or mean Average Precision (mAP) [1,2].

3. The authors argue that their approach is in the style of Sabb et al., but Saab et al. did not claim they solved a localization problem.

4. In this study, the author verified the model using the 3D saliency maps, but do not qualify them using the overlap with ground truth bounding boxes. Adebayo et al. [3] find that "reliance, solely on visual assessment can be misleading" and that "visual inspection is a poor guide in judging whether an explanation is sensitive to the underlying model and data."

Minor:

- 1. It is exaggerated to say that the framework is related to model distillation and cross-modal learning
- 2. Figure 3b caption says, "a fully supervised model trained using n= 400 manually annotated training example". At line 527, it says "391 exams from 235 patients are hand-labeled". The authors should check the numbers.

[1] Kuo W, Häne C, Mukherjee P, Malik J, Yuh EL. Expert-level detection of acute intracranial hemorrhage on head computed tomography using deep learning. *Proceedings of the National Academy of Sciences*. 2019 Nov 5;116(45):22737-45.

[2] Navarro F, Sekuboyina A, Waldmannstetter D, Peeken JC, Combs SE, Menze BH. Deep Reinforcement Learning for Organ Localization in CT. *arXiv preprint arXiv:2005.04974*. 2020 May 11.

[3] Adebayo J, Gilmer J, Muelly M, Goodfellow I, Hardt M, Kim B. Sanity checks for saliency maps. In *Advances in Neural Information Processing Systems 2018* (pp. 9505-9515).

Reviewer #3:

Remarks to the Author:

The authors performed an extensive revision in which they addressed most of the comments in a reasonable way. I have no further comments.

Reviewer #4:

Remarks to the Author:

We thank the authors for the revised version of the manuscript and for addressing the criticisms raised. Overall, the manuscript has been substantially improved and the proposed methodology will be a great contribution to the field of automated imaging screening. However, the article could be improved by addressing the following comments:

- 1) Please reconsider the statements about using PET/CT data for palliative care decisions. This is completely dependent on diagnosis, available therapies, patient preferences, and advances in medical science. The PET/CT data would not play a more significant role than so many other variables, e.g., age, comorbidity, cancer diagnosis, and cancer stage. Likely the latter are simple and just as important features as this presented PET/CT data abstraction model. This is not to say that such PET/CT models will never be a component of this decision making, but this is too premature and too speculative at this moment and not strongly supported by the data.
- 2) We appreciate that more details are added about how mortality is measured. However, the authors do not explain from what time point mortality is measured. Is it from the first available scan until last available scan or death, whichever comes first? How is mortality calculated in a person without a date of death and with only one scan in that case? The authors should also consider adding a statement about the censoring mechanism. If patients are censored at the last scan, the censoring time will likely not be independent of the mortality risk. That is, patients with multiple scans and therefore longer time to censoring compared to patients with only one scan, may tend to be more ill and hence have an elevated mortality risk.
- 3) In the x-day mortality prediction models, the authors should strongly consider including also the censored patients as the authors are now only fitting and evaluating performance in patients where date of death is known, which is information that is not known at the time of the scan. One way to handle censoring when evaluating x-day mortality/survival is by using inverse probability of censoring weighting.
- 4) The comparison between a simple mortality prediction model and the same model also including the output from the scan model could have been done for x-day mortality instead of using Cox models. The authors should consider this to reduce the number of analyses in the manuscript and enhance readability.
- 5) The authors use a concordance statistic, but do not write which one or how it was estimated? Was it Harrells' C?
- 6) The authors compare nested Cox models using a likelihood ratio test approach. However, a model can be significantly better than another, but not show any substantial improvement in predictive performance. Therefore, we suggest that the authors do not focus on significant differences in model fit between various Cox models or logistic regression models (if considering x-day mortality prediction), but instead only focus on out-of-sample predictive power.
- 7) In the Cox models displayed in Supplementary Table 16 and 17, the authors include only patients with one of the most frequent indications in the training data. Is that correct? If so, which indication is considered the reference group, i.e., the indication which all other indications are compared against. This information may be useful to make it more clear which indications are included. The same argument can be made about the abnormalities in Supplementary Table 16 and 18.
- 8) In supplementary table 17, the authors write "Log hazard ratios for single variable Cox models fit on each covariate and a multivariable Cox model fit on all covariates" in the caption. However, the table does not seem to contain results from both univariable and multivariable analyses.

Responses to Reviewer #1

R1.1 Thanks to the authors for kindly attempting to address some of my concerns. The writing and clarity have improved vastly, but I am still concerned about this paper’s novelty and the performance of the framework.

We appreciate the reviewer’s positive feedback about writing and clarity, and again express our thanks to the reviewer for their feedback, which has helped us to improve the manuscript.

We respectfully disagree with the reviewer’s concerns about novelty and performance. As we describe in detail below, we believe our approach represents a significant departure from previous work on ML for full-body volumetric imaging. The performance we achieve provides compelling evidence that this approach could provide value in a clinical context. The analyses presented in this manuscript are the first of their kind and would be of use to practitioners and researchers interested in machine learning for large volumetric imaging.

R1.2 In addition, I am not sure which concept(s) is (are) proved when the authors claim this work is a proof-of-concept: (i) The concept of using NLP to generate weakly labels and then training a DL model for medical image analysis is not new; (ii) The concept that multi-task learning outperforms single-task learning has been extensively proved; and (iii) the authors did not solve an abnormality “localization” problem (see below).

We thank the reviewer for raising this question, as it gives us the opportunity to clarify the positioning of this work. We believe that our approach is novel along both clinical and technical axes.

To the best of our knowledge, existing literature presents no other automated abnormality detection models that are trained on the full dimensionality of FDG-PET/CT scans. For example, the model of Sibille *et al.* accepts as input small segments of the full volume each with dimensionality $9 \times 64 \times 64$ [11]. The input to our model is a full scan measuring $300 \times 256 \times 256$, which contains more than 500 times the number of voxels. Training models on the smaller inputs is easier, but it requires manually-segmenting the training data. In contrast, because our model accepts the full scan, it can be trained on weak labels extracted automatically from the report. This approach represents a major departure from the way others before us have trained models for full-body volumetric images. Our manuscript is the first to provide evidence that this approach can be used to derive clinically compelling insights from PET/CT, and given the large size of each scan, the nuanced language of PET/CT reports, and the complex structure of the output space, it was far from obvious that such an approach should work prior to this study.

The manuscript proposes a series of technical design choices that make this approach viable. (i) The language of PET/CT reports is complex and varies greatly from patient

to patient, so instead of relying on a rule-based systems to extract labels, we use BERT, a state of the art natural language processing algorithm (note that a paper focused just on the idea of using BERT for chest x-ray reports was published only one month ago at EMNLP [58]). (ii) To be compatible with these weak labels extracted at the exam-level, our model must accept the full scan as input. With such a high-dimensional input space, it can be hard to avoid overfitting if we only have a few positive examples. We identify this problem and show that a location-based multi-task learning formulation is critical for achieving strong performance on rare abnormalities. Note that multi-task learning does not always outperform single-task learning; whether it does is highly dependent upon the specific multi-task formulation. One of the technical contributions of this paper is the demonstration that the specific location-based multi-task formulation we propose does actually improve performance compared to single-task learning. Additionally, to our knowledge, we are the first to show that multi-task attention mechanism we propose can be effective in isolating the regions of the whole-body PET/CT scan that are relevant for each task.

It is the integrated application of concepts (i) and (ii) listed by the reviewer that produces a result for item (iii). The reviewer questions the novelty of each contribution individually; we clarify that our main innovation rests not only in the individual contributions described above – which we maintain are substantial – but also in their combination. Together our contributions make it possible to weakly supervise models that accept full volumetric scans as input and provide high levels of performance: an approach to training not previously explored on PET/CT or similarly sized scans. Our manuscript identifies the challenges that arise when we try to apply weak supervision to these large PET/CT scans and explores design choices for overcoming them. Such an analysis does not exist in the current literature and we believe that our manuscript would be of immense value to those interested in training machine learning models on large volumetric images without manually segmenting them. (Note that our response to **R1.4** addresses the reviewer’s comment on item (iii)).

R1.3 The authors argue that there is no great deal of practical difference between an AUC of 0.8 vs. 0.75, because in this study, “very high sensitivity is required, but nearly any level of specificity would be helpful.” This is not true. Both sensitivity and specificity are equally important; otherwise, the authors cannot rank positive examples more highly than negative examples in Fig 2b-d. In other words, the authors provided an analysis of the models’ AUC (not only sensitivity) in Figure 2b-d. Please refer to <https://developers.google.com/machine-learning/crash-course/classification/roc-and-auc> for the interpretation of AUC.

We appreciate the reviewer’s comment regarding model performance. First, we would push back on the notion that both sensitivity and specificity are equally important. In a screening setting, sensitivity is commonly more important than specificity: a drop in specificity means a radiologist may have to interpret a few more exams whereas a drop in sensitivity means a patient with the disease will be misdiagnosed. In this setting, and in most others, false positives and false negatives do not have the same consequences.

Because of this, we argue that models with AUCs of 0.8 and 0.75 can be quite useful, since we can choose a threshold to optimize sensitivity or specificity, whichever is more important. This is done in the recent work of Titano et al. (Nature Medicine), which describes a 3D-CNN volumetric image classifier that achieves an AUC of 0.73 on gold-standard labels in the task of identifying acute neurologic events in head CT [52]. Ultimately, we believe that it depends on the use case and the threshold selected for each; Titano et al. claim that a triage system could be constructed with their work. We make clear how usable thresholds could be selected in the context of our work by visualizing sensitivity curves in Figures 2b-d.

The sensitivity curves displayed are meant to demonstrate how many total exams must be interpreted by the radiologist in order to achieve near 100% sensitivity to abnormal exams. ROC curves, if used in a similar fashion, would demonstrate the proportion of *normal* exams that must be read in order to achieve near 100% sensitivity to abnormal exams. While we agree that the two curves are highly related, they are fundamentally different. For example, the sensitivity curves, in conjunction with the oracle plots in grey, directly account for class imbalance, allowing the reader to more directly evaluate the impact our model would have to radiologist efficiency without requiring knowledge of class priors per region.

R1.4 The authors should not refer to the problem as abnormality “localization”. While the framework provides fine-grained labels indicating the localization of abnormality, it does not try to solve an object localization problem. As I have pointed out in my previous review, to localize abnormality in PET/CT, the framework needs to generate pixel-level bounding boxes of abnormal findings. The authors should conduct evaluations by comparing generated boxes with gold references and report standard metrics such as intersection over union ratio (IoU) or mean Average Precision (mAP) [1,2].

As the reviewer points out, our model does predict the “localization of abnormality”, so in the previous manuscript we described the problem as “abnormality localization”. We agree with the reviewer that the term “object localization” in the computer vision community means generating bounding boxes around the object. To avoid confusion, we’ve modified our description of the clinical task to “anatomically-resolved abnormality detection” and describe our model as doing “abnormality detection and location estimation.” This change is reflected in the text, and a selected subset of the changes can be found below for convenience.

While reading an FDG-PET/CT scan, a radiologist will typically record the anatomical regions with abnormal metabolic activity within a multi-paragraph report. This task, which we call ~~abnormality localization~~ *anatomically-resolved abnormality detection*, is clinically important in FDG-PET/CT studies. However, manually labeling a dataset for ~~localization~~ *anatomically-resolved abnormality detection* is particularly painstaking because it requires either performing pixel-level annotations or sorting abnormalities into a hierarchy of anatomical regions [10].

...

To address these challenges, we present a machine learning framework for training ~~abnormality localization~~ abnormality detection and location estimation models for large medical images without manual labeling or segmentation.

...

~~Localization~~ Estimating abnormality location, from a modeling standpoint, is key in that it mitigates hidden stratification, or performance variability in subgroups of data due to imprecise labels [5]. In our work, we pursue abnormality ~~localization~~ location estimation in part to provide explicit supervision over many regions of the body, in contrast to our single-task models trained on summary codes describing the whole body.

R1.5 The authors argue that their approach is in the style of Sabb et al., but Saab et al. did not claim they solved a localization problem.

We reference Saab et al.’s broader training procedure, which mirrors our own through its use of labels derived from unstructured text reports to train weakly supervised image classifiers. Additionally, and perhaps more relevantly, we reference the authors’ claim that their “model predictions appropriately reflect the spatial localization of [intracranial hemorrhage], which was investigated via a channel-wise occlusion procedure.”

R1.6 In this study, the author verified the model using the 3D saliency maps, but do not qualify them using the overlap with ground truth bounding boxes. Adebayo et al. [3] find that “reliance, solely on visual assessment can be misleading” and that “visual inspection is a poor guide in judging whether an explanation is sensitive to the underlying model and data.”

We thank the reviewer for their feedback regarding our use of saliency maps. We are largely in agreement with the reviewer that it would be desirable to compare directly with ground truth bounding boxes; unfortunately, we do not have the ground truth bounding boxes needed to conduct a meaningful evaluation, as these are not provided in clinical practice and would be enormously expensive to create. In the paper, we emphasize that we display the 3D saliency maps only to make the point that the task heads are roughly aligned with their respective regions of interest. In light of the reviewer’s comments, we have further softened the language related to the 3D saliency maps.

Analysis and visualization of attention distributions, such as those shown in Figure 5b-c, can be paired with analysis of two-dimensional and three-dimensional saliency maps (shown in Figure 5d-g and Fig 2e, respectively) in order to ~~sanity check~~ assess model behavior. ~~This said, the reader should note that these tools merely offer a first step towards coarse-grained model~~

interpretation. Future work should explore whether or not they can be leveraged to reliably produce fine-grained segmentations of abnormalities.

R1.7 It is exaggerated to say that the framework is related to model distillation and cross-modal learning

We thank the reviewer for the input. We have updated the related work section to include more work that is directly relevant to our framework, and removed the reference to model distillation.

However, we point out that cross-modal machine learning is often used when one modality is used to supervise another, which is exactly what is done here. See *e.g.* Dunmon *et al.* "Cross Modal Data Programming Enables Rapid Medical Machine Learning", which uses cross-modal learning in exactly this manner [24]. Thus, we have retained these citations.

R1.8 Figure 3b caption says, "a fully supervised model trained using n= 400 manually annotated training example". At line 527, it says "391 exams from 235 patients are hand-labeled". The authors should check the numbers.

We thank the reviewer for catching this error. The figure and its caption has been updated.

We compare three models: (yellow, left) a fully supervised model trained using ~~n=400~~ $n = 391$ manually annotated training examples, (blue, middle) a weakly supervised model trained using $n = 6,530$ automatically annotated training examples, and (green, right) a hybrid model trained using a combined dataset of ~~n=6,930~~ $n = 6,921$ training examples, ~~n=400~~ 391 of which are manually annotated.

Responses to Reviewer #3

R3.1 The authors performed an extensive revision in which they addressed most of the comments in a reasonable way. I have no further comments.

We thank reviewer for their comment and for their detailed feedback in the previous round of revisions.

Responses to Reviewer #4

R4.1 We thank the authors for the revised version of the manuscript and for addressing the criticisms raised. Overall, the manuscript has been substantially improved and the proposed methodology will be a great contribution to the field of automated imaging screening. However, the article could be improved by addressing the following comments:

We thank the reviewer for the positive feedback and detailed suggestions for improvements.

R4.2 Please reconsider the statements about using PET/CT data for palliative care decisions. This is completely dependent on diagnosis, available therapies, patient preferences, and advances in medical science. The PET/CT data would not play a more significant role than so many other variables, e.g., age, comorbidity, cancer diagnosis, and cancer stage. Likely the latter are simple and just as important features as this presented PET/CT data abstraction model. This is not to say that such PET/CT models will never be a component of this decision making, but this is too premature and too speculative at this moment and not strongly supported by the data.

We agree with the reviewer and appreciate this feedback. Palliative care decisions are extremely complex and depend on a whole host of variables some not easily measured quantitatively. Our intention in training mortality prediction models from the scan alone was to validate that the representation our model learned is useful for other tasks of clinical interests and to show the potential utility of PET/CT data in palliative care decisions. We do show that PET/CT data can provide additional signal not present in more easily-attained covariates. However, we did not intend to argue that our models are ready for use by palliative care teams. We agree with the reviewer that some of the language in the previous version of the manuscript could suggest the latter and have made changes to address this.

We demonstrate empirically that our multi-task representation is critical for strong performance on rare abnormalities with limited training data. ~~and for accurately predicting mortality~~ The representation also contributes to more accurate mortality prediction from imaging data (AUROC 80%), ~~which could assist palliative care teams~~ suggesting the potential utility of our framework beyond abnormality detection and location estimation.

(See line 10 to line 15)

Our framework enables ~~abnormality localization~~ anatomically-resolved abnormality detection in FDG-PET/CT scans, and models based on our framework could potentially provide clinical value in ~~two~~ three ways: (1) by automatically screening negatives and enabling the prioritization of exams and (2) by enabling future diagnostic aids that could help radiologists localize abnormalities within the scan, flag rare abnormalities and quickly draft reports. ~~and (3) by yielding a learned representation of each scan that can enable more accurate mortality predictions, which is important for palliative care planning.~~

(See line 60 to line 67)

Specifically, we find that our 3D-convolutional neural network, when pre-trained on multi-task anatomically-resolved abnormality ~~localization detection~~, can be fine-tuned to predict mortality within a ~~90~~180-day period

(AUROC 8076%), a critical challenge for hospitals looking to identify patients that could benefit from palliative care.

(See line 110 to line 113)

One of these tasks, mortality prediction, is of immense importance in palliative care. The National Palliative Care registry estimates that less than half of all patients that need palliative care actually receive it [16]. Palliative care involves holistic and nuanced decision-making that depends on a number of hard-to-measure variables including patient wishes, diagnosis and available therapies. While it is beyond the scope of this study to explore how automatic interpretation of imaging data might fit into this complex decision making process, our framework addresses some of the technical obstacles to leveraging automated analysis of imaging data in palliative care workflows. Statistical models that integrate the abnormality predictions we propose in this work alongside traditional covariates could eventually play a quantitative role in efficiently identifying patients who may benefit most from palliative care, and who might have otherwise been missed by current care models. In the context of our work, both identifying abnormalities in a triage setting and better flagging patients who are likely to die within clinically important time frames (e.g. 90 days) may represent new capabilities that could improve patient care.

(See line 388 to line 400)

We chose to train mortality prediction models on imaging data alone not because doing so would necessarily be optimal in clinical practice, but because it highlights some of the technical challenges associated with integrating high-dimensional imaging data into mortality prediction. It demonstrates the utility of this data source in mortality prediction.

(See line 457 to line 461)

R4.3 We appreciate that more details are added about how mortality is measured. However, the authors do not explain from what time point mortality is measured. Is it from the first available scan until last available scan or death, whichever comes first? How is mortality calculated in a person without a date of death and with only one scan in that case? The authors should also consider adding a statement about the censoring mechanism. If patients are censored at the last scan, the censoring time will likely not be independent of the mortality risk. That is, patients with multiple scans and therefore longer time to censoring compared to patients with only one scan, may tend to be more ill and hence have an elevated mortality risk.

The reviewer raises good points and we have rerun our analysis in response. In the revised manuscript, patient's without a date of death in the Social Security Death Index are assumed alive and are censored on the date we accessed the data (November 4, 2019). A patient's mortality is measured from their last scan until their death or the

accession date, whichever comes first. This design is described in detail in the revised methods:

For mortality prediction and survival analysis, we use the same dataset of 4,926 patients that we use for the abnormality localization detection experiments. Dates of death for the cohort were retrieved using STARR, which integrates medical records at Stanford with the Social Security Death Index (SSDI). Of the patients in the dataset, 867 (18%) had a date of death recorded in STARR. Patients were not prospectively followed for clinical outcomes in this retrospective study design. Patients without a date of death in the SSDI are still included in training, validation, and testing. For these patients we assume survival and censor at the date we accessed the data (November 4, 2019).

(See line 780 to line 787)

To evaluate the capacity of our scan model to predict mortality from PET/CT imaging data alone, we frame mortality prediction as a binary classification task: given a patient's last PET/CT scan in the dataset, predict whether the patient's date of death falls within x days of the scan date.

(See line 788 to line 791)

In this survival analysis, the response variable is the number of days between the patient's last PET/CT exam and their date of death. ~~Unlike the binary mortality prediction experiments described above,~~ For patients without a date of death in the SSID, we assume survival and right censor on the ~~date of their last PET/CT exam in our data.~~ we accessed the data.

(See line 801 to line 805)

R4.4 In the x -day mortality prediction models, the authors should strongly consider including also the censored patients as the authors are now only fitting and evaluating performance in patients where date of death is known, which is information that is not known at the time of the scan. One way to handle censoring when evaluating x -day mortality/survival is by using inverse probability of censoring weighting.

We agree that censored patients should be included in the training and evaluation of the deep x -day mortality prediction models. As we describe in our methods, we have updated our analysis to incorporate this feedback. Now, for both training and evaluating we include all patients and assume that those who did not have a date of death recorded in the Social Security Death Index were alive at the time we accessed the data (November 4, 2019). Because the latest scan in our dataset was performed in 2010, this means we assume all patients without a date of death are alive 365 days after their scan. We have rerun all of our x -day mortality prediction experiments with this new inclusion criteria and updated the results. Our findings have not changed substantially from a qualitative standpoint. We do still observe a statistically significant

increase in performance due to multi-task pretraining, but now observe this increase when $x = 180$ days instead of $x = 90$ days (note that we apply the Bonferroni correction when interpreting these p-values to account for multiple hypothesis testing). See the relevant changes below:

We updated Figure 4d.

Our multi-task model predicts mortality within ~~90~~180 days with an AUROC of ~~79.9% (76.2%, 83.8%)~~ 75.8% (73.5%, 78.1%). At this threshold, multi-task pre-training enables a ~~15.3%~~ 14.07% improvement in mean AUROC over pre-training on Kinetics (~~69.3% to 79.9%, $p = 0.0271$ paired permutation test~~) (66.5% to 75.8%, $p = 0.0031$ paired permutation test). Similarly, compared to pretraining on FDG-PET/CT summary codes in a single-task setting (see *Methods*), multi-task pre-training enables a ~~15.3%~~ 14.6% improvement in predicting mortality within ~~90~~180 days (~~69.3% to 79.9%, $p = 0.0226$ paired permutation test~~) (66.2% to 75.8%, $p = 0.0077$ paired permutation test). Figure 4d compares our model to the two baselines at each of the four thresholds.

(See line 288 to line 295)

We thank the reviewer for the suggestion of using IPCW for handling the censoring issue. Since our data includes no patients censored within 365 days (our highest threshold for x -day mortality prediction), we do not utilize IPCW here.

R4.5 The comparison between a simple mortality prediction model and the same model also including the output from the scan model could have been done for x -day mortality instead of using Cox models. The authors should consider this to reduce the number of analyses in the manuscript and enhance readability.

We thank the reviewer for the suggestion. We performed the suggested comparison for x -day mortality. At all four thresholds $x = 45, 90, 180,$ and 365 the logistic regression model with the output from the scan model achieves higher test AUROC than the model without. For example, when $x = 90$, the regression with output from the scan model achieves an AUROC of 84.5% while regression without achieves an AUROC of 61.4%. We have added these results to Supplementary Table 20.

While we could have replaced our Cox models with these x -day logistic regression models in the main text, we chose not to for a few reasons: (1) discretizing the response variable throws out some potentially relevant variability in time to death among patients, (2) with x -day mortality we have to compare AUROC at four different thresholds while with Cox models we can simply compare Harrell's concordance index which summarizes model performance with a single metric, and (3) the changes discussed in **R4.4** unify the censoring mechanisms of the x -day mortality and the Cox models, which we think improves the readability of this section while allowing us to keep the Cox models.

R4.6 The authors use a concordance statistic, but do not write which one or how it was estimated? Was it Harrell's C?

We used Harrell's concordance index. We've clarified this in the methods and added a citation.

Furthermore, to estimate out-of-sample predictive power, we fit a Cox model on the validation set and compute Harrell's concordance index on the test set using the implementation provided by lifelines [44,47]
(See line 859 to line 861)

R4.7 The authors compare nested Cox models using a likelihood ratio test approach. However, a model can be significantly better than another, but not show any substantial improvement in predictive performance. Therefore, we suggest that the authors do not focus on significant differences in model fit between various Cox models or logistic regression models (if considering x-day mortality prediction), but instead only focus on out-of-sample predictive power.

We agree with the reviewer that out-of-sample predictive power is the more important metric to report here, so in the updated results section we now mention it before the likelihood ratio test. Note that we saw substantial improvements in out-of-sample predictive performance.

We compare the out-of-sample predictive power of these Cox models by fitting on the validation set and computing Harrell's concordance index on the test set. A Cox model fit just on age, summary code and indication achieves a concordance index of 0.6090.597, whereas a model that incorporates abnormality location prediction achieves a concordance index of 0.7290.636. Via the likelihood ratio test, we also show that the fit with abnormality predictions is significantly better than the fit without ($p = 3.7 \times 10^{-11}$) ($p = 7.4 \times 10^{-5}$) [21]. Indeed, the locations of hypermetabolic abnormalities seem to provide useful signal not present in other, more simply-attained covariates such as age or indication.
(See line 313 to line 321)

While it doesn't measure predictive performance, the likelihood ratio test does show that the output of the scan model can explain additional variance in the outcome that the baseline covariates (e.g. age and indication) cannot. We think this is an important finding and decided to keep a sentence in the results mentioning it.

R4.8 In the Cox models displayed in Supplementary Table 16 and 17, the authors include only patients with one of the most frequent indications in the training data. Is that correct? If so, which indication is considered the reference group, i.e., the indication which all other indications are compared against. This information may be useful to make it more clear which indications are included. The same argument can be made about the abnormalities in Supplementary Table 16 and 18.

We include all of the patients in all of our cox models. This is an important clarification and we thank the reviewer for pointing it out. We have updated our methods to more clearly describe (1) which patients are included in fitting the cox models and (2) how we encoded the indications:

We fit Cox proportional hazard models on all patients that were not used to train the scan model (*i.e.* the patients in the validation and test sets). When computing out of sample predictive power, we fit on the validation set and evaluate on the test set.

(See line 806 to line 809)

Of the 42 different indications in our dataset, we consider the 13 with more than two occurrences among patients with a recorded date of death in the test set. These 13 indications are: breast cancer restaging, cervical cancer, colorectal cancer, head-neck cancer, lung cancer, lymphoma, ovaries, diagnostic breast cancer, diagnostic lung cancer, diagnostic head-neck cancer, regional or whole body, diagnosis lymphoma, and uncovered scan. Each patient's indication is encoded using 13 binary variables taking on the value 0 (without the indication) or 1 (with the indication). A patient with none of these 13 indications would be assigned a value of 0 for all 13 variables. In the Cox models, the 0 is used as reference group for each indication.

(See line 818 to line 825)

Unlike the categorical indication variable, the abnormality location estimates are not binary variables but rather continuous variables representing probabilities.

R4.9 In supplementary table 17, the authors write Log hazard ratios for single variable Cox models fit on each covariate and a multivariable Cox model fit on all covariates in the caption. However, the table does not seem to contain results from both univariable and multivariable analyses.

We thank the reviewer for pointing out this typo. We have made the following correction in the manuscript: Log hazard ratios for a multivariable Cox model fit on indication, patient age and summary code. ~~Log hazard ratios for single variable Cox models fit on each covariate and a multivariable Cox model fit on all covariates~~. Shown are Wald 95% confidence intervals and p values.

Reviewers' Comments:

Reviewer #1:

Remarks to the Author:

The authors have addressed most of my comments. I have not further comments.

Reviewer #4:

Remarks to the Author:

We thank the authors for responding thoroughly to our concerns. We have no further comments.

Responses to Reviewer #1

R1.1 The authors have addressed most of my comments. I have not further comments.

We thank the reviewer for their detailed comments throughout the review process.

Responses to Reviewer #4

R4.1 We thank the authors for responding thoroughly to our concerns. We have no further comments.

We thank the reviewers for their thorough comments and suggestions on the survival analysis in our work.